Manuscript prepared for Ocean Sci.
with version 5.0 of the LATEX class copernicus.cls.
Date: 4 November 2022

# Quantifying the impacts of the Three Gorges Dam on the spatial-temporal water level dynamics in the upper Yangtze River estuary

Huayang Cai[1,2], Hao Yang[1,2], Pascal Matte[3], Haidong Pan[4], Zhan Hu[5], Tongtiegang Zhao[6], and Guangliang Liu[7]

[1]Institute of Estuarine and Coastal Research/State and Local Joint Engineering Laboratory of Estuarine Hydraulic Technology, School of Ocean Engineering and Technology, Sun Yat-sen University, Guangzhou, 510275, China
[2]Guangdong Provincial Engineering Research Center of Coasts, Islands and Reefs/Southern Marine Science and Engineering Guangdong Laboratory (Zhuhai), Zhuhai, 519082, China
[3]Meteorological Research Division, Environment and Climate Change Canada, Quebec, QC G1J 0C3, Canada
[4]First Institute of Oceanography, and Key Laboratory of Marine Science and Numerical Modeling, Ministry of Natural Resources, Qingdao, 266061, China
[5]School of Marine Sciences, Sun Yat-sen University, Zhuhai, 519082, China
[6]School of Civil Engineering, Sun Yat-sen University, Zhuhai, 519082, China
[7]Shandong Provincial Key Laboratory of Computer Networks, Qilu University of Technology (Shandong Academy of Sciences), Jinan, 250353, China

*Correspondence to:* Guangliang Liu (guangliangliu@163.com)

**Abstract.** Understanding the alterations in spatial-temporal water level dynamics caused by natural and anthropogenic changes is essential for water resources management in estuaries, as this can directly impact the estuarine morphology, sediment transport, salinity intrusion, navigation conditions, and other factors. Here, we propose a simple triple linear regression model linking the water level variation on a daily timescale to the hydrodynamics at both ends of an estuary. The model was applied to the upper Yangtze River estuary (YRE) for examining the influence of the world's largest dam, the Three Gorges Dam (TGD), on the spatial-temporal water level dynamics within the estuary. It is shown that the regression model can accurately reproduce the water level dynamics in the upper YRE, with a root mean squared error (RMSE) of 0.061-0.150 m seen at five gauging stations for both the pre- and post-TGD periods. This confirms the hypothesis that the response of water level dynamics to hydrodynamics at both ends is mostly linear in the upper YRE. The regression model calibrated during the pre-TGD period was used to reconstruct the water level dynamics that would have occurred in absence of the TGD's freshwater regulation. Results show that the spatial-temporal alterations in water levels during the post-TGD period are mainly driven by the variation in freshwater discharge due to the regulation of the TGD, which results in increased discharge during the dry season (from December to March) and a dramatic reduction in discharge during the wet-to-dry transitional period. The presented method to quantify the separate contributions made by changes in boundary conditions and geometry on spatial-temporal water level dynamics is particularly useful

for determining scientific strategies for sustainable water resources management in dam-controlled or climate-driven estuaries worldwide.

## 1 Introduction

Water level is an important factor affecting estuarine environments as they influence hydrological, ecological, and biogeochemical processes in many ways (such as flood control, water quality, carbons and nutrients cycles). It has previously been demonstrated that water level dynamics are mainly controlled by river flow alteration in the catchment and tidal variation near the estuary mouth, resulting in a positive surface water level gradient along the estuary axis in the landward direction (Buschman et al., 2009; Sassi and Hoitink, 2013). However, the relationship between water level dynamics and hydrodynamics at both ends of an estuary may be impacted by anthropogenic interventions (such as dam construction, channel dredging, or land reclamation). Hence, quantifying the water level dynamics in artificially modified environments is essential for understanding hydrological regime shifts and improving the sustainable management of water resources in estuaries.

Water level dynamics in estuaries are nonstationary since they are subject to nonlinear interactions with the barotropic tide that can be modified by channel geometry, bottom friction, and river discharge. This nonlinear relationship can be approximated by the balance between tidally averaged residual water level slope and bottom friction. As a consequence, the water level dynamics can be expressed by semi-analytical solutions of the one-dimensional St. Venant equations, provided that adequate information (tidal forcing at the estuary mouth, river discharge at the upstream end, and simplified channel geometry) is available (e.g., Cai et al., 2014a,b, 2016, 2019a; Kästner et al., 2019). However, semi-analytical solutions can only capture the first-order hydrodynamics due to the fact that they usually require simplifications of the topography (e.g., rectangular or exponential cross-sections) and flow characteristics (e.g., small Froude number, predominant $M_2$ tide). Alternatively, enhanced harmonic analysis considering nonlinear and nonstationary tide-river interactions have been introduced to reproduce the spatial-temporal water level dynamics in estuaries with substantial freshwater discharge (e.g., Kukulka and Jay, 2003; Matte et al., 2013, 2014; Pan et al., 2018a,b; Gan et al., 2019; Guo et al., 2020). Despite their ability to predict water levels on a finer temporal scale (e.g., hourly), these methods suggest that water level dynamics in estuaries are highly nonlinear and nonstationary owing to complex tide-river interactions. In this study, we show that when the dynamics are examined at a coarser temporal resolution (e.g., daily averaged), the water level dynamics in some river estuaries may display a regular and predictable pattern which can be described as a first-order approximation by a relatively simple linear law.

Numerous studies have been conducted to understand the potential environmental impacts of the Three Gorges Dam (TGD), the largest dam in the world, since its operation beginning in 2003 has dramatically changed the downstream hydrology and sediment delivery in the Yangtze River. Key

factors influenced by the operation of TGD include hydrodynamics (Cai et al., 2019c), morpholog-
ical evolution (e.g., Yang et al., 2011, 2014; Lai et al., 2017; Yuan et al., 2020), sediment and flow
discharges (e.g., Chen et al., 2016; Guo et al., 2018), nutrient transport (e.g., Wang et al., 2020),
river-lake interaction (e.g., Guo et al., 2012; Mei et al., 2015), and thermal dynamics (e.g., Cai et al.,
2018a; Liu et al., 2018). However, due to the long distance from the TGD to the downstream estuary,
quantification of the potential impacts of the TGD (mainly due to its seasonal freshwater regulation)
on the spatial-temporal water level dynamics is a challenging task, as flow alterations are gener-
ally concurrent with geometric changes induced by natural and anthropogenic factors. In addition,
water level dynamics in the downstream estuary is highly sensitive to even small changes in the
upstream basin. Here, we present a simple yet powerful triple linear regression model linking the
water level variation at a daily timescale to hydrodynamics at both ends of the upper Yangtze River
estuary (YRE). The advantage of this regression model is that it allows a separate quantification
of the contributions made by changes in the boundary conditions and geometry, which are the two
most significant controlling factors for determining the water level dynamics. We test our regression
model on the observed water levels in the upper YRE to quantify the influence of the TGD on the
downstream spatial-temporal water level dynamics.

## 2   Study domain and datasets

### 2.1   Overview of the YRE

The Yangtze River, which flows from west to east in central China, is one of the world's most
important rivers due to its great economic and social relevance. It has a length of about 6300 km and
a basin area of about 190,000 km$^2$ (Figure 1a). The Yangtze River basin is geographically divided
into four parts, the upper, central, lower sub-basins, and an estuary area, and has connections at
Yichang, Jiujiang, and Datong (DT) hydrological stations (Figure 1a). Of particular concern in this
study is the impact of the TGD, the world's largest dam, on the spatial-temporal patterns of tide-river
dynamics in the downstream estuary. It is located about 45 km upstream of Yichang (Figure 1a). The
TGD project began in 2003; by 2009, when full operation began, the total water storage capacity rose
to ~40 km$^3$, equivalent to 5% of the Yangtze's annual discharge. Downstream of Datong, where the
upstream tidal limit is located, the YRE extends  630 km to the seaward end of the South Branch.
Wuhu (WH), Maanshan (MAS), Nanjing (NJ), Zhenjiang (ZJ), Jiangyin (JY), and Tianshenggang
(TSG) are major gauging stations along the mainstream in the seaward direction (Figure 1b). The
river discharge shows distinct seasonal patterns due to the controlling effect of the Asian monsoon
on the region's climate. For example, from 1979-2014, more than 70% of freshwater discharge at
DT occurred during the wet season (May-October).

Apart from river flows, upstream propagating tides are also a major source of hydrodynamic
energy in the upper YRE, which is characterized by a meso-tide with a mean tidal range of ~2.7

m near the estuary mouth at Zhongjun station. According to observations at the Zhongjun station, the average ebb tide duration (7.4 h) is longer than the averaged flood tide duration (5 h), indicating an irregular semidiurnal character (Zhang et al., 2012). Unlike previous studies (e.g., Qiu and Zhu, 2013; Lu et al., 2015; Alebregtse and de Swart, 2016) which focused on tidal hydrodynamics near the estuary mouth, here, we mainly concentrate on the water level dynamics under the impacts of the TGD's seasonal regulation over the upper reach of the YRE.

## 2.2  Datasets

Hydrological data for both the pre-TGD (1978-1984) and post-TGD (2003-2014) periods of water level from six tidal gauging stations mentioned above along the estuary were collected, together with the corresponding river discharges observed at the DT hydrological station. Here, it is worth noting that the observed river discharges at the DT hydrological station were generally derived from well-calibrated stage-discharge relationship, which is established by concurrent measurements of stage and discharge (through approximately 50-70 filed measurements of flow depth and velocity in each year to account for the cross section changes) over a wide range of river discharge conditions. These data were obtained from the Yangtze Hydrology Bureau of the People's Republic of China. The daily averaged water levels were determined by averaging the hourly values, which were interpolated from daily high and low water levels using shape-preserving piecewise cubic interpolation. All the water levels at different gauging stations were corrected to the national mean sea level of Huanghai 1985. The data during the period 1985-2002 was not included since most of the water level data were not available. However, the collected data were sufficient to represent the hydrodynamic condition before and after the TGD's operation.

## 3  Method

### 3.1  Triple linear regression model

In this study, we hypothesize that the water level dynamics on a daily time scale shows a regular and predictable pattern. Thus, we propose that the daily-mean water level variation $Z$ (at an arbitrary location within the estuary) in response to hydrodynamics observed at both ends of the estuary can be described by the following triple linear regression model:

$$Z = Z_0 + \alpha Q/\mathrm{std}(Q) + \beta Z_{\mathrm{down}}/\mathrm{std}(Z_{\mathrm{down}}) + \gamma Z_{\mathrm{up}}/\mathrm{std}(Z_{\mathrm{up}}). \tag{1}$$

Here, $Z_0$ is the intercept representing a base water level which is in equilibrium with climate and local conditions, so that the water level variation is linearly proportional to the river discharge $Q$ imposed at the upstream boundary, and the water levels $Z_{\mathrm{down}}$ and $Z_{\mathrm{up}}$ are imposed at the seaward and upstream boundaries of the estuary, respectively. In Equation (1), 'std' denotes the standard devia-

tion. Here, the seaward boundary should be in principle located far from the upstream boundary with negligible river discharge influence. In this study, the DT hydrological station was chosen as the upstream end, while the TSG gauging station was used as the downstream end. The source code of the proposed triple linear regression model is available at https://github.com/Huayangcai/Triple-Linear-Regression-Model-V1.0-Matlab-Toolbox. It is worth noting that there is no unique stage-discharge relationship at the DT hydrological station (see Figure S1 in the Supplementary Material) owing to the stage-discharge hysteresis effect caused by flow unsteadiness, together with the influence of external forcing, either the potential influence induced by the tidal forcing (especially during the dry season) or the exerted residual water level slope upstream of the DT hydrological station (owing to the relative importance of river discharge between the main stream and the tributaries, especially during the flood season). Thus, in order to explicitly account for the influence of external forcing in both upstream and downstream reaches, here we have explicitly introduced the $z_{\mathrm{up}}$ into the regression model, and hence the dynamics of residual water level slope along the upper YRE. $Z_0$, $\alpha$, $\beta$ and $\gamma$ are linear regression coefficients that are determined from the observed data according to a least-squares fit technique. It should be noted that the imposed downstream water level $Z_{\mathrm{down}}$ also implicitly accounts for other nontidal factors, such as wind, ocean temperature and ocean salinity, which are assumed to be negligible in the regression model when compared with the tidally induced water level fluctuations featured by a typical spring-neap cycle (see Figure S2 in the Supplementary Material). In Equation (1), the relative importance of variance contributions made by riverine $p_{\mathrm{r}}$ and tidal $p_{\mathrm{t}}$ forcing can be estimated by the following formulas:

$$p_{\mathrm{r}} = \mathrm{var}\left[\alpha Q/\mathrm{std}(Q) + \gamma Z_{\mathrm{up}}/\mathrm{std}(Z_{\mathrm{up}})\right]/\mathrm{var}\left[\alpha Q/\mathrm{std}(Q) + \beta Z_{\mathrm{down}}/\mathrm{std}(Z_{\mathrm{down}}) + \gamma Z_{\mathrm{up}}/\mathrm{std}(Z_{\mathrm{up}})\right],$$
(2)

$$p_{\mathrm{t}} = \mathrm{var}\left[\beta Z_{\mathrm{down}}/\mathrm{std}(Z_{\mathrm{down}})\right]/\mathrm{var}\left[\alpha Q/\mathrm{std}(Q) + \beta Z_{\mathrm{down}}/\mathrm{std}(Z_{\mathrm{down}}) + \gamma Z_{\mathrm{up}}/\mathrm{std}(Z_{\mathrm{up}})\right] = 1 - p_{\mathrm{r}},$$
(3)

where 'var' denotes the variance.

### 3.2 Quantifying the separate impacts due to boundary and geometry changes

In order to quantify the geometric change induced by the combined influences of both natural and anthropogenic modifications and separate these from boundary effects (induced by the changes in upstream and downstream conditions, primarily due to the TGD's freshwater regulation), the entire study period is divided into two periods: pre-TGD and post-TGD. The data during the pre-TGD period is used for model calibration. Subsequently, the calibrated regression coefficients were then adopted for the same model over the post-TGD period to estimate the expected water levels if there

existed no significant geometric change induced by the construction of the TGD. Here we use the true observed hydrodynamics at both ends of the estuary (i.e., the discharge and water level at the upstream end and the open-ocean water level at the seaward end).

In this manner, the total alteration of water level (induced by both the boundary changes and the geometric alteration) in the post-TGD period relative to the pre-TGD period can be quantified as:

$$\Delta_{\mathrm{TOT}} = Z_{\mathrm{obs,post-TGD}} - Z_{\mathrm{obs,pre-TGD}}, \tag{4}$$

which represents the difference in observed water level for the post-TGD ($Z_{\mathrm{obs,post-TGD}}$) period and the pre-TGD ($Z_{\mathrm{obs,pre-TGD}}$) period. This total alteration is due to two distinct effects:

1) The contribution made by changes in the boundary conditions ($\Delta_{\mathrm{BOU}}$), defined as the difference between the water level values simulated for the post-TGD ($Z_{\mathrm{sim,post-TGD}}$) and pre-TGD ($Z_{\mathrm{sim,pre-TGD}}$) period:

$$\Delta_{\mathrm{BOU}} = Z_{\mathrm{sim,post-TGD}} - Z_{\mathrm{sim,pre-TGD}}. \tag{5}$$

2) The contribution made by changes in the geometry ($\Delta_{\mathrm{GEO}}$), defined as the difference between the observed ($Z_{\mathrm{obs,post-TGD}}$) and simulated ($Z_{\mathrm{sim,post-TGD}}$) values of water level for the post-TGD period:

$$\Delta_{\mathrm{GEO}} = Z_{\mathrm{obs,post-TGD}} - Z_{\mathrm{sim,post-TGD}}. \tag{6}$$

Equations (4)-(6) can be combined, yielding the following expression:

$$\Delta_{\mathrm{GEO}} = \Delta_{TOT} - \Delta_{\mathrm{BOU}} - \varepsilon, \tag{7}$$

where $\varepsilon = Z_{\mathrm{sim,pre-TGD}} - Z_{\mathrm{obs,pre-TGD}}$ represents the model bias (i.e., mean error) between the simulated and observed water level during the calibration period (i.e., the pre-TGD period). To evaluate the model performance in estimating water level alterations, we require that the bias $\varepsilon$ should be small when compared with $\Delta_{\mathrm{BOU}}$ and $\Delta_{\mathrm{GEO}}$ at different time scales (i.e., seasonal and annual).

It is worth noting that the quantity $\Delta_{\mathrm{BOU}}$ (including both the upstream and downstream boundary conditions) should be interpreted as the water level alteration owing to the overall influences driven by both human interventions and climate change. However, in this study the largest contribution to the alteration in upstream boundary condition (i.e., river discharge) can be primarily attributed to the TGD's operation, since the TGD alone accounts for more than 30% of the total storage capacity of the dams constructed between 1987 and 2014 along the Yangtze River (Li et al., 2016). In addition,

we note that the only other dam (Gezhouba, abbreviated by GZB, see Figure 1a) along the main

course of the Yangtze River was constructed in 1981 (before the TGD) and should not considerably

influence the discharge regime since it is a run-of-the-river hydroelectric system. With regard to

the downstream boundary condition, the adopted water levels observed at TSG station implicitly

account for the potential impacts induced by both anthropogenic (such as channel dredging) and

climate (such as global sea level rise) changes. Meanwhile, it is also worth noting that the quantity

$\Delta_{\mathrm{GEO}}$ should be interpreted as the water level alteration due to the overall impacts caused by both

the bathymetric change and the storage area change.

## 4    Results

### 4.1    Performance of the triple linear regression model

The proposed triple linear regression model was applied to reproduce the water level dynamics

observed during both the pre-TGD and post-TGD periods for the given upstream river discharges

and water levels observed at the DT hydrological station and the water levels observed at the TSG

gauging station (see Figure 2). The values of the three regression coefficients and the intercept

were determined by the least squares method taken between the observed and predicted daily water

levels. The model performance was then evaluated in terms of the value of the root mean square

error (RMSE). It can be seen from Figure 2 that our model can satisfactorily reproduce the water

level dynamics along the upper YRE, with an RMSE that ranges from 0.061-0.150 m (4%-13% of

the standard deviations of the observed water levels, see Table 1) at the five water level stations,

which leads support to our hypothesis that the response of water level dynamics to hydrodynamics

at both ends of the estuary is largely linear in the upper YRE owing to the explicit inclusion of $Z_{\mathrm{up}}$

in the regression model. Table 1 presents the calibrated linear regression coefficients for both study

periods, where we observe a general reduction in the $Z_0$, $\alpha$ and $\beta$ parameters, and an increase in

the $\gamma$ parameter, after the construction of the TGD. To clarify the importance of including $Z_{\mathrm{up}}$ in

the regression model, we replaced the terms $\alpha Q/\mathrm{std}(Q) + \gamma Z_{\mathrm{up}}/\mathrm{std}(Z_{\mathrm{up}})$ with the nonlinear term

$\alpha[Q/\mathrm{std}(Q)]^{\beta}$ in Equation (1). In this case, the model performance is more or less the same as the

original triple linear regression model (see Figure S3 and Table S1 in the Supplementary Material),

but the RMSE values are slightly larger at NJ, MAS and WH stations (ranging between 0.17 and

0.21 m) than those using the triple linear regression model (ranging between 0.11 and 0.15 m).

     Spatial interpolation of the triple linear regression coefficients was performed by means of piece-

wise cubic Hermite interpolants (e.g., Matte et al., 2014) in order to correctly reproduce the water

level dynamics at arbitrary locations along the estuary. Figure 3 shows the four spatially interpo-

lated model coefficients together with vertical error bar along the upper YRE for the pre-TGD and

post-TGD periods. Generally, a longitudinal reduction in coefficients (e.g., $Z_0$ and $\beta$ in Figure 3a,

c) in the landward direction suggests a weakening effect of these parameters on the total variations

in water levels, which corresponds to the external forcing from the seaward end of the estuary. On the contrary, if the coefficients are increasing (e.g., $\alpha$ and $\gamma$ in Figure 3b, d), this corresponds to an enhancement from the upstream end. However, we observed an exception from the MAS to WH stations, where the coefficient $\alpha$ was reduced (see Figure 3b), suggesting a switch of the effect of river discharge in the upstream part of the estuary. The error bars presented in Figure 3 represent the standard error of the estimated linear regression coefficients, which suggests that the proposed triple linear regression model is fitting well.

## 4.2 Reconstructions of spatial-temporal water level dynamics

Using the calibrated regression models and interpolated linear regression coefficients (see Figure 3), the spatial-temporal water level dynamics for the two study periods can be reconstructed along the upper YRE for the climatological reference year (Figure 4), which is defined by evaluating for each day of the year the average value of all measurements available over the study period for the same day (though February $29^{th}$ during leap years was not considered). Subsequently, we used the Matlab 'gradient.m' function (i.e., 'gradient' calculates the central difference for interior data points, while it calculates values along the edges of the matrix with single-sided differences, see details in https://www.mathworks.com/help/matlab/ref/gradient.html) to estimate the residual water level slope based on the reconstructed water levels along the YRE. In Figure 4, we note that there is a local minimum water level slope which occurs in the central part (between JY and ZJ) of the YRE, which shifts by approximately 30 km landward after the TGD begins operation. Such a shift of local minimum water level slope is very likely to be linked to the abnormal tidal range reduction observed at the ZJ gauging station after the TGD begins operation (Cai et al., 2019c) and this might be related to a minimum in energy flux divergence (Giese and Jay, 1989; Jay et al., 2015), with implications for sedimentary processes.

Figure 5 shows comparisons of the longitudinal variation of the water levels and their slopes during the four seasons. It can be observed that the most significant changes in these two parameters occurs in autumn and winter seasons, which correspond to a dramatic reduction in river discharge during the wet-to-dry transition period (i.e. autumn) and slightly increased river discharge during the dry season (i.e. winter) due to the operation of the TGD since 2003. Conversely, changes during the spring and summer are relatively minor, which is mainly due to negligible change in the river discharge. It should be noted that the water levels in the downstream reaches ($x < 200$ km) were slightly increased during the spring, while they are approximately constant in the upstream part. However, caution should be taken with the interpretation of levels and slopes resulting from regression coefficients interpolated by splines: cubic splines might be regarded as over-fitting but smooth quadratic splines were found to introduce greater spatial undulations in levels and slopes.

### 4.3 Influence of the TGD on the spatial-temporal water level dynamics

Using Equations (4)-(7), the triple linear regression model can quantify the contributions induced by the changes in boundary conditions (i.e., upstream freshwater and water level alterations at DT and downstream water level alteration at TSG) and in geometry to the water level variability during the post-TGD period. In this study, the regression model calibrated during the pre-TGD period was successively applied to the post-TGD period, keeping the same coefficients (i.e., $Z_0$, $\alpha$, $\beta$, $\gamma$) obtained

before. The simulated water levels were compared with the actual measurements and their differences (i.e., $\Delta_{\mathrm{GEO}}$ in Equation (4)) represent the alterations caused by geometric changes, which can be attributed to the combined influences of natural and anthropogenic changes. Compared to the pre-TGD period, it is possible to isolate the influence on water level dynamics from the boundary conditions impacts (i.e., $\Delta_{\mathrm{BOU}}$ in Equation (3)).

Table 2 presents monthly averaged and annual alterations of water levels during the post-TGD period calculated from Equations (4)-(7) based on the observed and simulated water levels for the pre- and post-TGD periods. It can be seen that the model bias $\varepsilon$ is generally smaller than the calculated $\Delta_{\mathrm{BOU}}$ and $\Delta_{\mathrm{GEO}}$ (with $\varepsilon/\Delta_{\mathrm{BOU}}$ and $\varepsilon/\Delta_{\mathrm{GEO}}$ being 0.8% and 0.1% at the annual scale on average, respectively), which suggests that the impacts due to model errors on the analysis of

water level dynamics is negligible. At the annual scale, we observe that the changes in the boundary conditions tends to increase the mean water level, while the geometric effect acts in the opposite direction, leading to an overall reduction in water level along the upper YRE.

Figure 6 shows the intra-annual variability (in a climatological year) of water level alterations at five gauging stations along the upper YRE. It is observed that the overall impacts of boundary

conditions and geometry effects can be divided into three distinct periods. In January to March, the total alteration $\Delta_{\mathrm{TOT}}$ averaged approximately 0.28 m over five different gauging stations along the upper YRE, while its average was small (0.01 m) during May to June and negative (approximately -0.54 m) on average for the rest of the year (see Figure 6a and Table 2). Noticeably, the increase of $\Delta_{\mathrm{TOT}}$ from January to March is mainly caused by changes in the boundary conditions (see Figure

6b), which is primarily attributed to the freshwater regulation of the TGD, and leads to an increased discharge during the dry season. Additionally, a significant decrease of $\Delta_{\mathrm{TOT}}$ in autumn (from September to November) is observed, due to the combined effects of boundary conditions and geometry. In Figure 6b, we observe that the alterations caused by boundary condition variations $\Delta_{\mathrm{BOU}}$ are positive throughout the year except for October and November, which can be primarily attributed

to the operation of the TGD, corresponding to a substantial reduction in freshwater discharge during the wet-to-dry transitional period. Such a boundary effect is partially due to the rise of the seaward water level, especially during the period when freshwater discharge is reduced (see Figure 7). The water level alteration caused by the geometric effect $\Delta_{\mathrm{GEO}}$ is negative and tends to increase along the channel, which is due to the cumulative effect of mean water level in the landward direction.

We now quantify the alterations in variance contributions made by riverine (denoted by $\Delta p_{\mathrm{r}}$) and

tidal (denoted by $\Delta p_{\mathrm{t}}$) forcing using Equations (2) and (3) to understand the impacts of freshwa-
ter regulation on the spatial-temporal water level dynamics. On average, it can be seen from Table
3 that the contributions made by the riverine forcing $p_{\mathrm{r}}$ to the overall water level variance are in-
creased during the post-TGD period. In particular, the $p_{\mathrm{r}}$ values at the JY and ZJ gauging stations
were substantially increased by 16.16% and 13.61%, respectively. Further upstream, less alteration
(ranging from 0.16%-1.87%) by the riverine forcing contributed to the overall water level variance.
Figure 8 displays the monthly alterations of the riverine and tidal contributions, which shows two
distinct types of responses, corresponding to the tide-dominated and river-dominated regions. At the
JY gauging station where the tide dominates over the river discharge, a larger alteration in $p_{\mathrm{r}}$ occurs
during the wet season, with two local maximum $\Delta p_{\mathrm{r}}$ values occurring in May and November, re-
spectively. Upstream from the ZJ gauging station where the river discharge dominates over the tide,
the alteration pattern of $p_{\mathrm{r}}$ is opposite to that in the tide-dominated region, with larger values occur-
ring during the dry season. It is worth noting that the local minimum water level slope highlighted
in Figure 5 coincides with the transition between the tide-dominated and river-dominated domains.
For detailed monthly averaged variance contributions made by riverine and tidal forcing during both
the pre- and post-TGD periods, the reader can refer to Figures S4-S5 in the Supplementary Material.
Here, it should be noted that the contribution $p_t$ implicitly accounts for both tidal and nontidal fac-
tors (e.g., wind, ocean temperature and ocean salinity), hence further study is required to quantify
the potential influences due to nontidal factors.

## 5 Conclusions

In this study, we have explored the alterations in spatial-temporal water level dynamics along the
main course of the YRE, with a special focus on quantifying the effects caused by the changes in
boundary conditions and geometry. Through the use of a triple linear regression model, we recon-
structed the spatial-temporal water level dynamics solely induced by changes in boundary conditions
in the post-TGD period. When compared to the observed and simulated values in the pre-TGD pe-
riod, it is possible to quantify the alterations attributed to the boundary conditions and geometry via
Equations (4)-(7). We show that the spatial-temporal alteration in water level dynamics is closely
related to the variation in freshwater discharge, which is mainly driven by the regulation of the TGD,
leading to an increased discharge during the dry season (from December to March) and a dramatic re-
duction in discharge during the wet-to-dry transitional period. Consequently, minor increases ($\sim$0.28
m) in water level are observed from January to March, while considerable decreases ($\sim$0.54 m) are
observed from July to December. The alterations induced by the variation of boundary conditions
are positive throughout the year except during October and November which showed a substantial
reduction of freshwater discharge owing to the TGD's operation. On the other hand, the alterations
caused by geometric changes are negative, which is mainly due to the riverbed deepening along the

channel.

It is notable that the alterations in water levels induced by the geometric changes $\Delta_{\mathrm{GEO}}$ (mainly caused by channel deepening) tend to increase in the landward direction (see Figure 6c). This phenomenon can be primarily attributed to the constant mean sea level or the ultimate base level that topography tends to approach due to erosion. This is illustrated by Figure 9, which shows the adjustment of the surface elevation profile to the change in bed profile, where we can observe an increase in the alteration of water level (i.e., $|\Delta Z| = |Z_0 - Z_1|$, where $Z_0$ and $Z_1$ represent the water levels for the original and new surface elevation profile) along the channel. In addition, this phenomenon is also closely related to the scouring downstream near the TGD, which slowly propagates further downstream due to the reduced sediment supply (see also Lamb et al., 2012; Sassi et al., 2012; Kästner et al., 2017). Moreover, the reduction of seasonal discharge variation due to TGD's regulation may probably reduce the overdeepening near the sea

Although the proposed triple linear regression model can satisfactorily reproduce the daily water level hydrodynamics along the upper YRE, the adopted boundary conditions at both ends of an estuary are not fully independent since the water level dynamics at TSG gauging station are influenced by the upstream river discharge observed at DT hydrological station, especially during the wet season which brings substantial freshwater discharge. Such a drawback can be improved by using water level dynamics, either observed or predicted using harmonic analysis, from an outer gauging station that has negligible impact from freshwater discharge. Our results here suggest that the construction of the TGD may have impacted the morphological evolution and hence the geometry in the estuarine area since the sediment loads observed at DT have decreased from 470.4 million tons annually in 1951-1985 to 138.7 million tons in 2003-2015, a substantial reduction of approximately 70% (Guo et al., 2018). However, it is difficult to separate the sediment trapping effect due to the TGD on geometric change from other natural and anthropogenic factors. It is also worth noting that in this study we assumed a more or less stationary condition before and after the TGD's construction for the regression model, which is not completely true due to the gradually increased geometric influence (such as ongoing scouring) caused by the TGD (e.g., Yang et al., 2022). In addition, it should be noted that the limited data length during the pre-TGD period may impact the modeling performance. However, even when using the limited data considered here, the proposed triple linear regression model can well reproduce the spatial-temporal water level dynamics and quantify the alterations made by changes in boundary conditions and geometry.

There exists a long tradition of statistical, analytical and numerical studies on tide-river interactions in estuaries worldwide, such as the Columbia River estuary in the USA (e.g., Kukulka and Jay, 2003; Jay et al., 1990; Pan et al., 2018b), the St. Lawrence River estuary in Canada (e.g., Godin, 1999; Matte et al., 2013, 2014), the Mahakam River estuary in Indonesia (e.g., Buschman et al., 2009; Sassi and Hoitink, 2013), the Yangtze River estuary in eastern China (e.g., Guo et al., 2015, 2020; Yu et al., 2020) and the Pearl River estuary in southern China (e.g., Zhang et al., 2018; Cai

et al., 2018b, 2019b). These studies showed that as tides propagate along the estuary the tidal amplitude, phase and shape were influenced by the bottom friction, channel geometry and river discharge. In this study, with the proposed simple yet effective triple linear regression model, we are able to isolate and to quantify the impacts of the boundary (such as freshwater regulation due to dam's operation) and geometric (such as channel dredging) effects on the tide-river dynamics. Such a novel approach should be particularly helpful for determining scientific guidelines for sustainable water resources management (e.g., dredging for navigation, flood control, salt intrusion prevention etc.) in estuaries worldwide, especially for dam-controlled estuaries. In addition, the proposed method can also be used to quantify the potential impacts of changes in boundary conditions induced by climate change (such as intensifying precipitation, global sea level rise, etc.) in natural estuaries without considerable human interventions.

### Data availability

The MATLAB codes and data used in this paper can be open-access from a publicly accessible and version-controlled GitHub repository (https://github.com/Huayangcai/Triple-Linear-Regression-Model-V1.0-Matlab-Toolbox).

### Author contributions

All authors contributed to the design and development of this work. The model was originally developed by HC. HY carried out the data analysis. GL built the model and wrote the paper. PM, HP, ZH and TZ reviewed the paper.

### Competing interests

The contact author has declared that neither they nor their co-authors have any competing interests.

### Financial support

This research has been supported by the National Natural Science Foundation of China (Grant No. 51979296), from the Guangdong Provincial Department of Science and Technology (Grant No. 2019ZT08G090), from the Guangzhou Science and Technology Program of China (Grant No. 202002030452).

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

505

**Table 1.** Calibrated linear regression coefficients for both the pre-TGD and post-TGD periods along the upper YRE

| | Stations | $Z_0$ | $\alpha$ | $\beta$ | $\gamma$ | RMSE/m | Standard deviation/m |
|---|---|---|---|---|---|---|---|
| JY | Pre-TGD | 1.82E-04 | 0.029 | 0.481 | 0.137 | 0.061 | 0.638 |
| | Post-TGD | -0.108 | -0.066 | 0.411 | 0.252 | 0.078 | 0.588 |
| ZJ | Pre-TGD | -0.041 | 0.402 | 0.432 | 0.422 | 0.114 | 1.233 |
| | Post-TGD | -0.129 | 0.125 | 0.365 | 0.657 | 0.120 | 1.123 |
| NJ | Pre-TGD | -0.211 | 0.478 | 0.312 | 0.957 | 0.128 | 1.725 |
| | Post-TGD | -0.409 | 0.209 | 0.289 | 1.066 | 0.145 | 1.541 |
| MAS | Pre-TGD | -0.202 | 0.475 | 0.259 | 1.317 | 0.135 | 2.031 |
| | Post-TGD | -0.385 | 0.305 | 0.240 | 1.280 | 0.150 | 1.804 |
| WH | Pre-TGD | -0.273 | 0.338 | 0.168 | 1.872 | 0.103 | 2.363 |
| | Post-TGD | -0.436 | 0.221 | 0.171 | 1.700 | 0.109 | 2.074 |

**Table 2.** Monthly averaged alteration in water level (m) attributed to changes in boundary condition ($\Delta_{\mathrm{BOU}}$) and to the geometry condition ($\Delta_{\mathrm{GEO}}$)

| Stations | Change | Jan | Feb | Mar | Apr | May | Jun | Jul | Aug | Sep | Oct | Nov | Dec | Annual |
|---|---|---|---|---|---|---|---|---|---|---|---|---|---|---|
| JY | $\Delta_{\mathrm{TOT}}$ | 0.11 | 0.16 | 0.10 | -0.06 | 0.07 | 0.05 | -0.02 | -2.39E-03 | -0.08 | -0.28 | -0.17 | 2.70E-03 | -0.01 |
| | $\Delta_{\mathrm{BOU}}$ | 0.26 | 0.31 | 0.26 | 0.12 | 0.29 | 0.29 | 0.21 | 0.23 | 0.16 | -0.07 | 0.02 | 0.16 | 0.19 |
| | $\Delta_{\mathrm{GEO}}$ | -0.16 | -0.17 | -0.16 | -0.17 | -0.22 | -0.25 | -0.23 | -0.24 | -0.24 | -0.20 | -0.17 | -0.15 | -0.20 |
| | $\varepsilon$ | 2.71E-03 | 0.02 | 1.32E-03 | 8.32E-04 | -3.96E-03 | 0.01 | -4.11E-04 | 0.01 | 7.03E-04 | -0.01 | -0.02 | -1.72E-03 | 1.01E-04 |
| ZJ | $\Delta_{\mathrm{TOT}}$ | 0.27 | 0.31 | 0.30 | -0.06 | 0.20 | 0.19 | 0.01 | 0.02 | -0.21 | -0.64 | -0.31 | 0.06 | 0.01 |
| | $\Delta_{\mathrm{BOU}}$ | 0.44 | 0.50 | 0.50 | 0.15 | 0.52 | 0.54 | 0.36 | 0.35 | 0.13 | -0.36 | -0.08 | 0.22 | 0.27 |
| | $\Delta_{\mathrm{GEO}}$ | -0.14 | -0.16 | -0.20 | -0.26 | -0.38 | -0.43 | -0.32 | -0.31 | -0.32 | -0.24 | -0.20 | -0.14 | -0.26 |
| | $\varepsilon$ | -0.04 | -0.02 | 2.87E-03 | 0.05 | 0.07 | 0.08 | -0.02 | -0.01 | -0.01 | -0.04 | -0.03 | -0.02 | 8.42E-05 |
| NJ | $\Delta_{\mathrm{TOT}}$ | 0.21 | 0.27 | 0.23 | -0.32 | 0.03 | 3.77E-03 | -0.27 | -0.27 | -0.58 | -1.22 | -0.70 | -0.15 | -0.23 |
| | $\Delta_{\mathrm{BOU}}$ | 0.57 | 0.62 | 0.67 | 0.15 | 0.66 | 0.71 | 0.48 | 0.43 | 0.09 | -0.63 | -0.22 | 0.23 | 0.31 |
| | $\Delta_{\mathrm{GEO}}$ | -0.30 | -0.32 | -0.45 | -0.56 | -0.73 | -0.80 | -0.71 | -0.68 | -0.66 | -0.53 | -0.45 | -0.34 | -0.54 |
| | $\varepsilon$ | -0.05 | -0.03 | 0.01 | 0.08 | 0.10 | 0.09 | -0.04 | -0.02 | -0.02 | -0.06 | -0.04 | -0.04 | 1.02E-04 |
| MAS | $\Delta_{\mathrm{TOT}}$ | 0.23 | 0.28 | 0.25 | -0.43 | -0.02 | -0.05 | -0.37 | -0.38 | -0.75 | -1.52 | -0.88 | -0.19 | -0.32 |
| | $\Delta_{\mathrm{BOU}}$ | 0.65 | 0.70 | 0.78 | 0.14 | 0.74 | 0.81 | 0.56 | 0.48 | 0.07 | -0.80 | -0.30 | 0.24 | 0.34 |
| | $\Delta_{\mathrm{GEO}}$ | -0.36 | -0.37 | -0.55 | -0.66 | -0.87 | -0.95 | -0.89 | -0.85 | -0.81 | -0.66 | -0.55 | -0.41 | -0.66 |
| | $\varepsilon$ | -0.06 | -0.05 | 0.02 | 0.09 | 0.10 | 0.09 | -0.04 | -0.02 | -0.02 | -0.07 | -0.03 | -0.02 | -6.67E-06 |
| WH | $\Delta_{\mathrm{TOT}}$ | 0.24 | 0.31 | 0.28 | -0.56 | -0.12 | -0.16 | -0.52 | -0.55 | -0.97 | -1.83 | -1.10 | -0.30 | -0.44 |
| | $\Delta_{\mathrm{BOU}}$ | 0.73 | 0.78 | 0.89 | 0.12 | 0.80 | 0.91 | 0.65 | 0.54 | 0.04 | -0.99 | -0.42 | 0.22 | 0.36 |
| | $\Delta_{\mathrm{GEO}}$ | -0.43 | -0.44 | -0.64 | -0.76 | -1.00 | -1.12 | -1.13 | -1.08 | -1.00 | -0.80 | -0.66 | -0.50 | -0.80 |
| | $\varepsilon$ | -0.05 | -0.04 | 0.02 | 0.07 | 0.07 | 0.05 | -0.03 | -0.01 | -0.01 | -0.04 | -0.02 | -0.02 | -1.30E-05 |

**Table 3.** Relative contributions made by riverine $p_r$ and tidal $p_t$ forcing for both the pre- and post-periods at annual scale

| Stations | $p_r$ (%) | | $p_t$ (%) | |
|---|---|---|---|---|
| | Pre-TGD | Post-TGD | Pre-TGD | Post-TGD |
| JY | 5.64 | 21.79 | 94.36 | 78.21 |
| ZJ | 54.65 | 68.27 | 45.35 | 31.73 |
| NJ | 87.89 | 89.75 | 12.11 | 10.25 |
| MAS | 94.69 | 94.86 | 5.31 | 5.14 |
| WH | 98.74 | 98.39 | 1.26 | 1.61 |

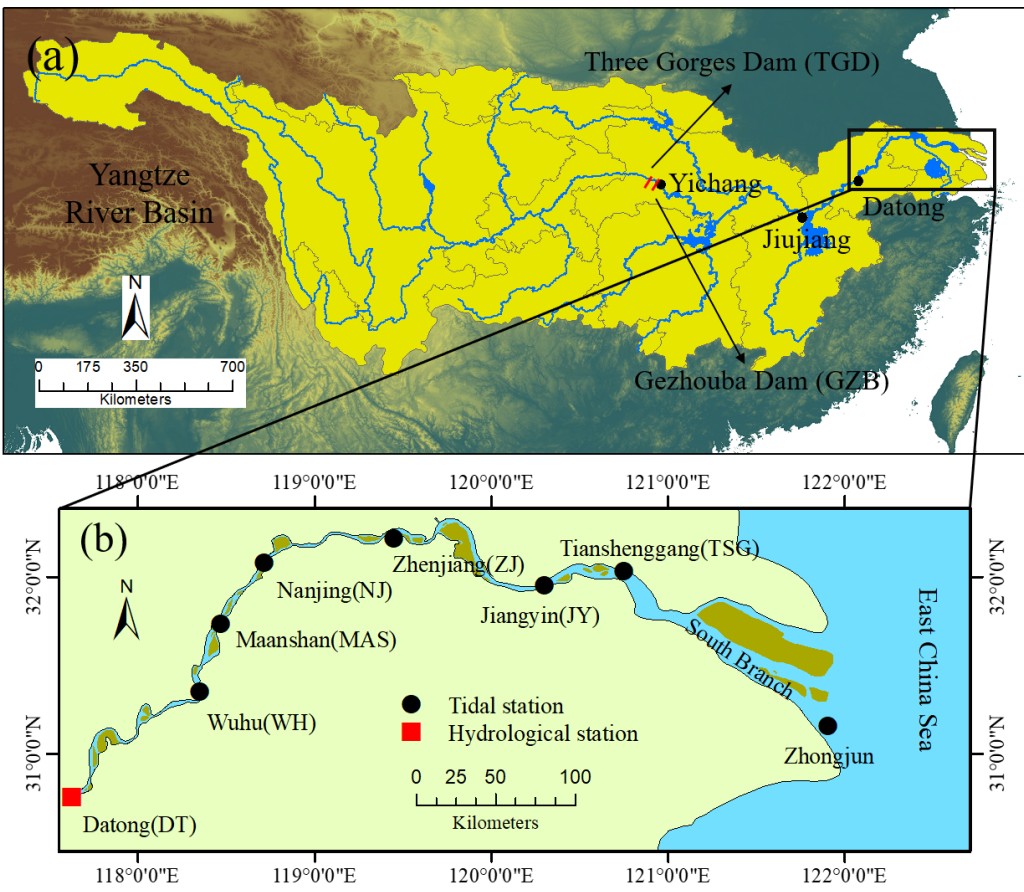

**Figure 1.** Map of the Yangtze River basin (a) and the YRE (b) displaying the observed tidal gauging stations and hydrological station.

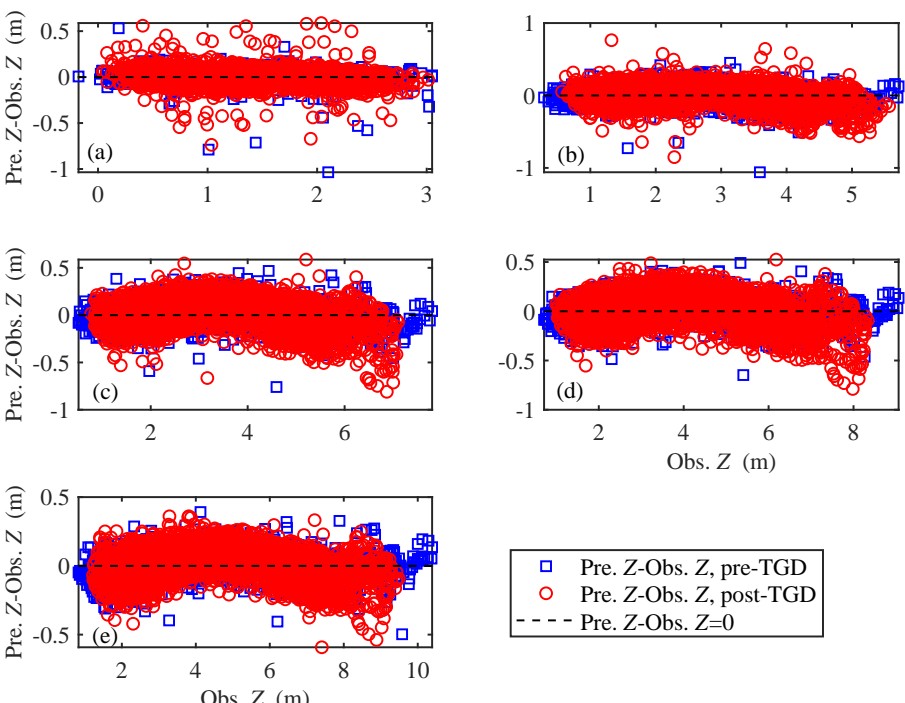

**Figure 2.** Alterations in difference between predicted and observed daily averaged water levels as a function of observed daily averaged water levels for both the pre-TGD and post-TGD periods at different gauging stations along the upper YRE: (a) Jiangyin (JY), (b) Zhenjiang (ZJ), (c) Nanjing (NJ), (d) Maanshan (MAS), (e) Wuhu (WH).

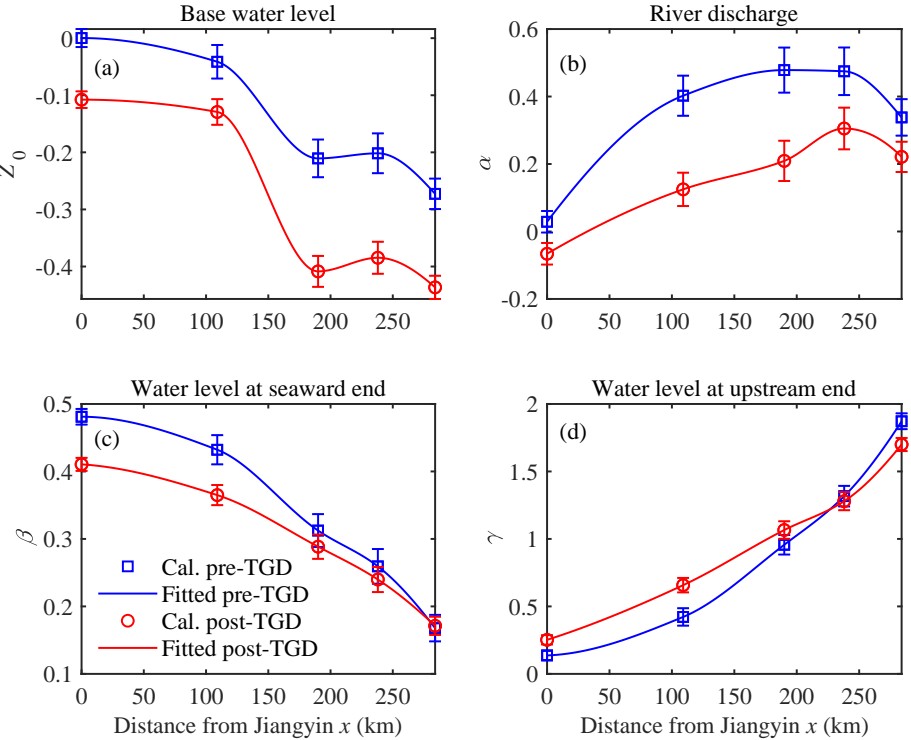

**Figure 3.** Interpolated linear regression coefficients $Z_0$ (a), $\alpha$ (b), $\beta$ (c), $\gamma$ (d) with error bar along the upper YRE (upstream of the Jiangyin gauging station) for both the pre-TGD and post-TGD periods. The vertical error bar was estimated using the Matlab 'regress.m' function with 95% confidence intervals.

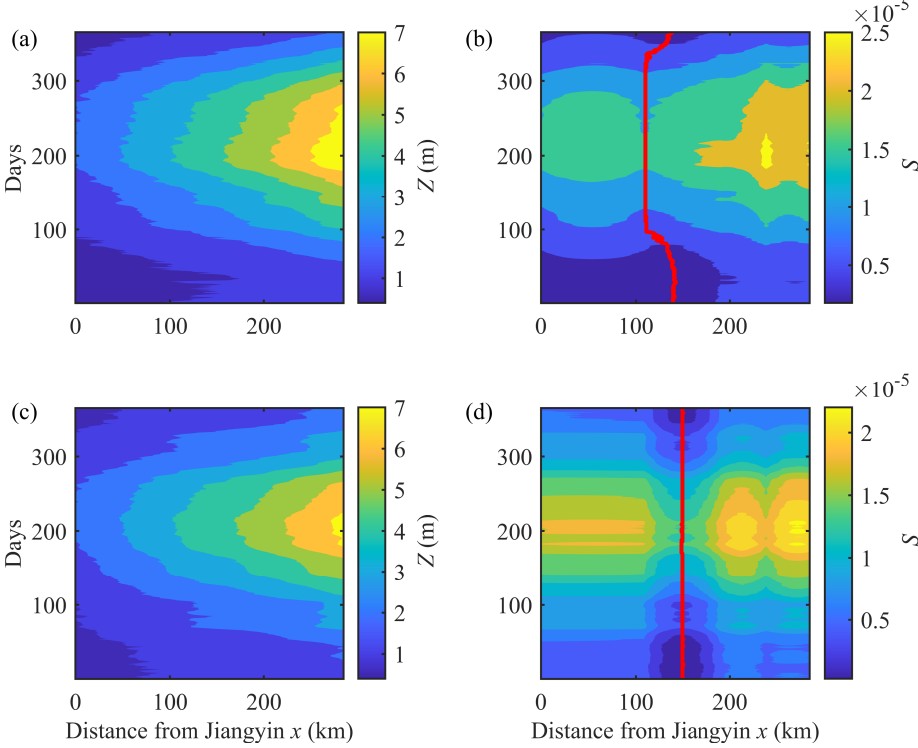

**Figure 4.** Reconstructed spatial-temporal water levels, $Z$, (a, c) and their slopes, $S$, (b, d) for the climatological year during both the pre-TGD (a, b) and post-TGD (c, d) periods. The red lines in subplots (b) and (d) indicate the local minimum water level slopes in the central section of the YRE (between Jiangyin and Zhenjiang).

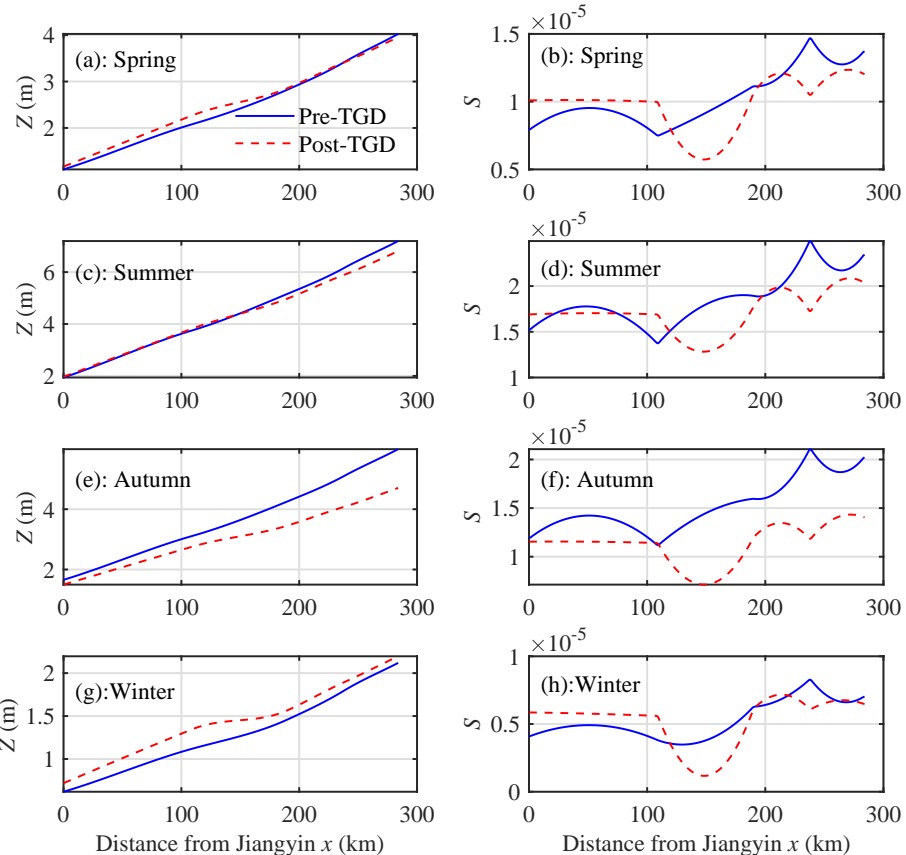

**Figure 5.** Longitudinal variability of reconstructed water level $Z$ (a, c, e, g) and its slope $S$ (b, d, g, h) along the upper YRE (from Jiangyin to Wuhu) during four seasons (spring: a, b; summer: c, d; autumn: e, g; winter: g, h) for the climatological year during the pre- and post-TGD periods.

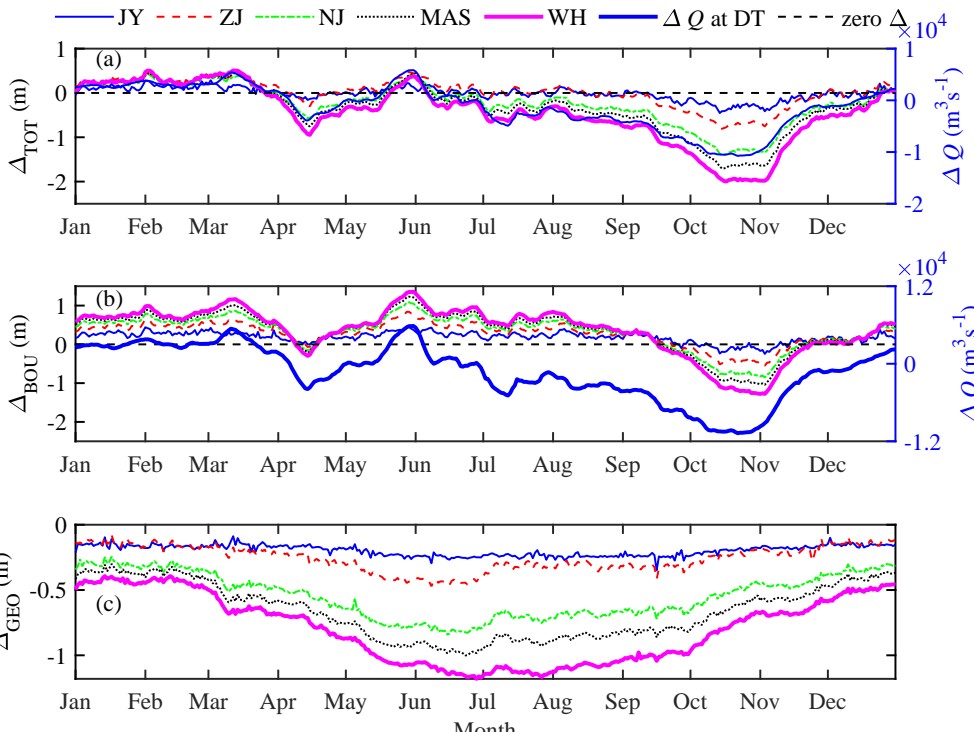

**Figure 6.** Alterations in water levels induced by the combined impacts of natural and anthropogenic changes $\Delta_{\mathrm{TOT}}$ (a), boundary condition changes $\Delta_{\mathrm{BOU}}$ (b), and geometric changes $\Delta_{\mathrm{GEO}}$ (c) at different gauging stations along the upper YRE.

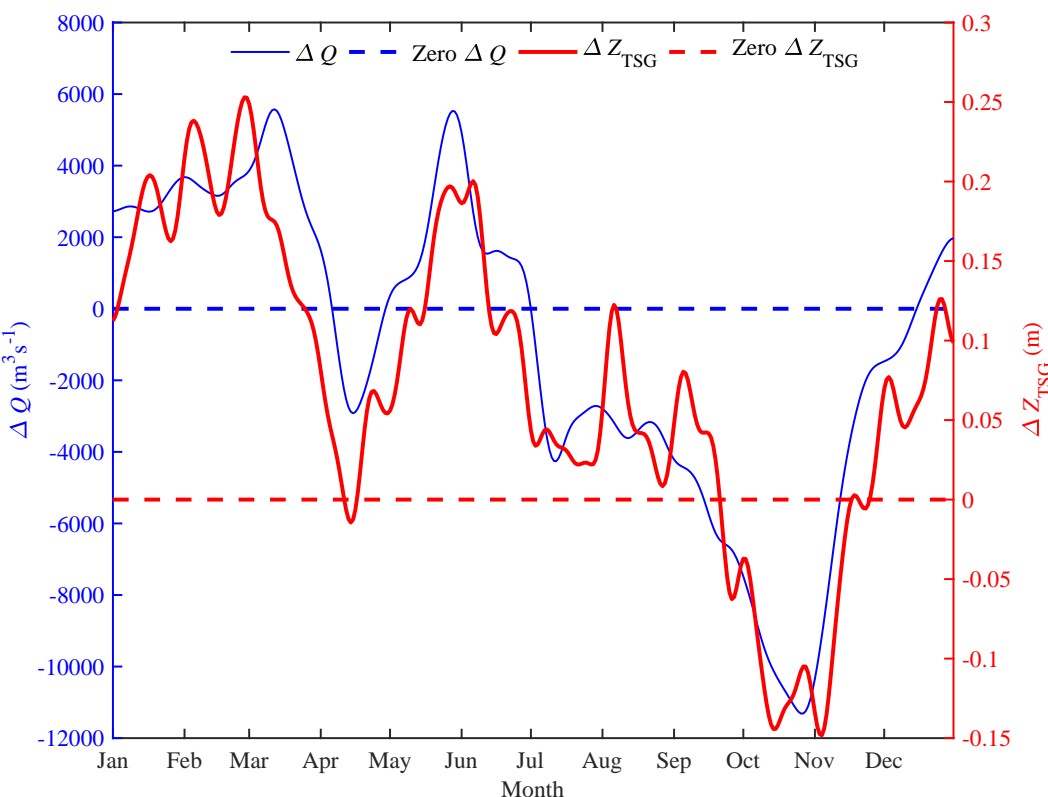

**Figure 7.** Alterations in river discharge and water level observed at DT and TSG, respectively, during the post-TGD period relative to the pre-TGD period over the climatological year. The daily averaged river discharge and water level were smoothed using a moving average filter with a span of 30 days.

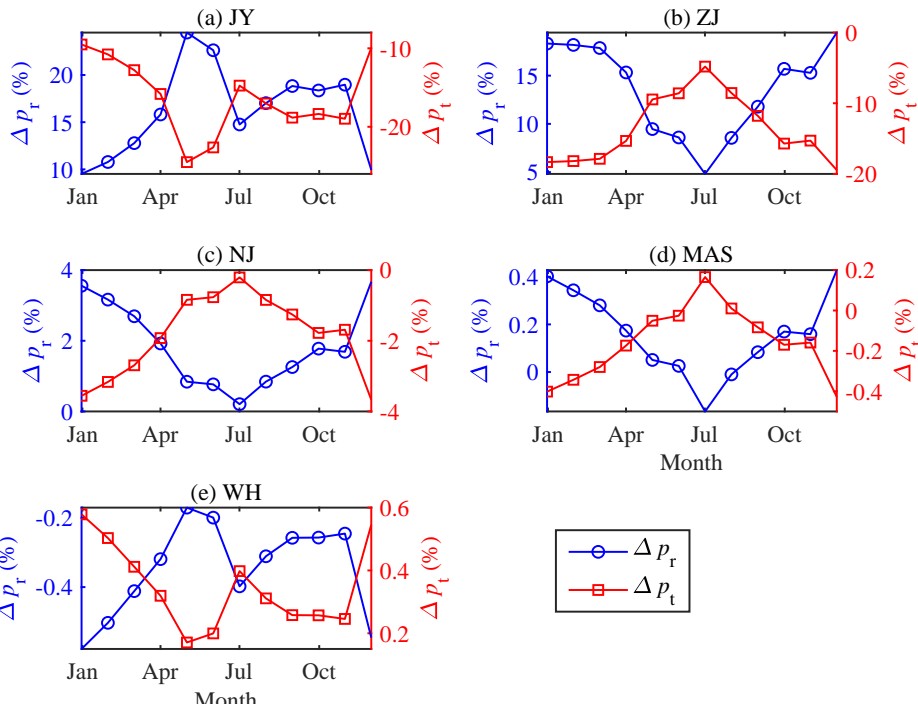

**Figure 8.** Alterations in variance contributions of riverine $\Delta p_\mathrm{r}$ and tidal $\Delta p_\mathrm{t}$ forcing at different gauging stations along the upper YRE: (a) Jiangyin (JY), (b) Zhenjiang (ZJ), (c) Nanjing (NJ), (d) Maanshan (MAS), (e) Wuhu (WH).

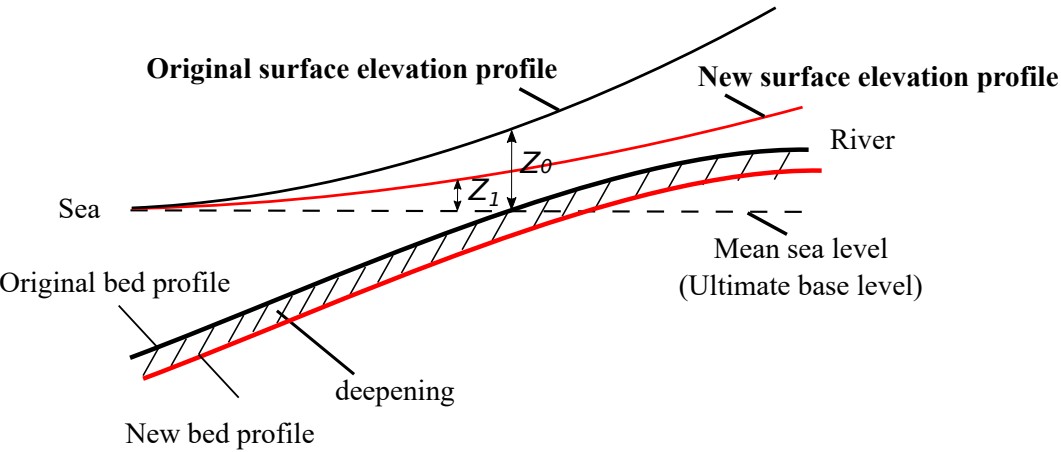

**Figure 9.** Illustration of the effect of riverbed deepening on the water level dynamics along the channel.