# Peer review of "Quantifying the impacts of the Three Gorges Dam on the spatial-temporal water level dynamics in the Yangtze River estuary"

_EGUsphere, 2022_

## Referee Comment (RC1)

Dear Editor Prof. Huthnance,

Thank you for sending me the manuscript: "Quantifying the impacts of the Three Gorges Dam on the spatial-temporal water level dynamics in the Yangtze River estuary" by Huayang Cai for review, which I read with great interest.

The authors apply a linear regression model to the tidally averaged water level in the Yangtze estuary to investigate the effects of the Three Gorges dam. The authors find that their regression model predicts the water level in the Yangtze reasonably well. They find that since construction of the dam, low flows have increased while flows during transition from the high to the low flow season have decreased.

The topic is very relevant and the manuscript was interesting to read. The applicability of a regression model to predict water levels in tidal rivers agrees with my own experience in this field. The text and figures are of high quality.

However, the regression model applied here is relatively simple, at least much simpler than previously applied models. This certainly makes it easy to grasp the results, especially for readers who are not experts on the topic. However, this also makes it difficult to identify the physical drivers behind changes in the water levels, and might introduce systematic errors. Below, I provide suggestions on how these issues can be verified and mitigated, if necessary.

Kind regards,

**Methods**

- The regression model includes both discharge and water level at the upstream station. As they depend on each other, the model is not parsimonious. As a consequence, the columns for $Q$ and $z_{\mathrm{up}}$ of the regression matrix will be close to collinear so that small changes (errors) in the data can result in large changes in the coefficients $\alpha$ and $\gamma$ even if the fit is good. Possible changes of the coefficients over time might thus be regression artefacts. This should be ruled out by verifying that $Q_{\mathrm{up}}$ and $z_{\mathrm{up}}$ are not strongly correlated.

  - If the correlation is weak, then the model is robust, but then it would be insightful to elaborate on why the upstream water level and discharge are unrelated. The comment "*influenced by the dynamics of [] tributaries*" (l. 119) is unclear. The water

level is uniquely determined by the backwater curve as long as the daily averaged water level does not change rapidly in time. Therefore, tributaries upstream of the inflow boundary influence downstream levels only through their discharge. Do the authors refer to tributaries downstream of the upstream station?

- If the correlation is strong, then it is better to replace the terms $\alpha\,Q + \gamma\,z_{\text{up}}$ with the non-linear term $a\,Q^b$. This model is less ambiguous. In my personal experience, the coefficients $a$ and $b$ of the non-linear model also give much more insight into the influence of the river discharge on the mean water level along tidal rivers.

- The regression model does not include the effect of the tides on the mean water level. However, this effect is not negligible during periods of low river flow (*LeBlond*, 1978). This introduces a systematic error. As the Three Gorges dam increased river discharge during the low flow season, this can bias the results. It is, therefore, reasonable to include the influence of tides on the mean water level in the regression model. For example, *Kukulka and Jay* (2003) suggest the regression model linear in $h^3$:

$$h^3 \approx a\,Q_{\text{river}}^2 + b\,|z_{\text{tide}}|^2 + c,$$

while (*Kästner et al.*, 2019) suggested linearizing the backwater equation, which can be readily approximated in a regression model linear in $h$ (or $z$).

- The independent variables are not normalized in the regression model so that the coefficients have very different magnitudes ($O(\alpha) = 10^{-5}$ while $O(\beta) = 1$). It is thus not obvious which predictor (downstream or upstream level) has the largest influence at a particular location. This can be revealed by normalizing the independent variables by their standard deviation before the regression:

$$z = z_0 + \alpha\,Q/\text{std}\,(Q) + \beta\,z_{\text{down}}/\text{std}\,(z_{\text{down}}) + \gamma\,z_{\text{up}}/\text{std}\,(z_{\text{up}})\,.$$

This is preferable to the order in the study, where variance is normalized after the regression.

- Interpolation of slopes and uncertainty estimates (Figure 4 and 5, lines 190ff)

- There is a mistake in the slope calculation. The values should be in the order $10^{-5}$, not $10^{-8}$. The distance between the stations was probably not converted from km to m.

- Determining the slope from by higher-order (Hermite) interpolation is not meaningful here. This is because the error (of the

slope) is amplified at the interpolated values between the stations. As a consequence, the interpolated slope has unrealistic local extrema at the midpoints between stations (Figure 5). The error of the cubically (Hermite) interpolated slope at the midpoint between stations is about 1.8 times as large as the errors of the levels at the stations. Since the error of the levels is about 10%, the error of the slope is about 20%. The local maxima of the slope, as well as the difference between the pre- and post-TGD period as indicated in Figure 5 are therefore insignificant. The interpolation error (of the slopes) can be considerably reduced by calculating the slopes at the midpoint between two stations and then linearly interpolating the slopes between the midpoints. In this case the error of the slope is only 0.7 times that of the error in levels. If the authors want to retain cubic interpolation, then the spurious extrema can be suppressed by fitting the 4 coefficients of the cubic polynomial with all 5 stations in a least squares manner.

– As the model is not parsimonious, it might fit well even if the regression coefficient are uncertain, as multiple parameter combinations can result in similar good model performance. A good way to assess the uncertainty is bootstrapping (*Efron and Tibshirani*, 1994). Simply split the time series into blocks comprising of one month, this reduces the effect of serial correlation. When there are $n$ blocks, randomly choose $\sqrt{n}$ blocks and fit the model. Repeat this a few hundred times. The standard error is simply the standard deviation of the estimated parameters. The standard error of the coefficient, predicted levels and slopes can then be indicated indicated with error bars in Figure 3 and 5. The cubic interpolation results in larger errors at midpoints between sections, so errors bars are best placed there.

**Minor**

Title estuary → upper estuary

35 The term "analytical solution" is misleading, as the water level is still determined by (numerically) integrating an initial value problem (eq. 22 in *Cai et al.* (2016)).

44 *Kukulka and Jay* (2003) should be referenced here, as an important regression model for the mean water level of tidal rivers.

45 *"these methods suggest that water level dynamics in estuaries are highly nonlinear and nonstationary"* This sounds as if water levels in tidal are

difficult to analyse and predict, and that looking at tidal cycle/average is a novel idea. However, there is a large amount of publications how water levels can be approximated well on a cycle-by-cycle basis, see the works of the groups of Savenije, Godin, Jay, Hoitink, and Friedrichs.

49 The reference to Darcy is dubious. Even if the surface level can be predicted by a linear regression model, it is still turbulent flow (quadratic flow resistance), which is very different from groundwater flow (linear flow resistance).

89 Mark Gaoqiaoju on the map in Figure 1

93 *"we mainly concentrate on the tide-river dynamics"* This is not the case, since, as commented by me before, the tidally induced water level offset is not included in the regression model.

110 Mention here, which of the stations where chosen as the upstream and the downstream end (Datong and Gaoqiaoju?).

169 *"linear"* is misleading here. The water depth in the upstream estuary most likely scale like $h \approx Q^{2/3}$. The non-linearity is just hidden by including $z_{\mathrm{up}}$ in the regression model.

248 The conclusion *"[at the downstream stations] tide dominates [the tidally averaged water leve]"* sounds odd, as the regression model applied in this study does not explicitly account for the tidally induced water level offset. It only includes the tidally averaged water level at the seaward station. However, at the river mouth the tidally induced water level offset is negligible as it integrates along the estuary, c.f. *Kästner et al.* (2019) and *Cai et al.* (2016). So, no meaningful conclusion about the tidal influence can be drawn. The model probably indicates that fluctuations of the sea level unrelated to tides, such as wind, ocean-temperature and ocean-salinity, dominate the mean water level dynamics near the sea. It would be insightful to actually determine the tidal influence by including it explicitly in the regression model.

248 The river discharge influences the salinity gradient, and with it the variation of the water level at the reference station at the sea (*Savenije*, 2012). The influence on river discharge on the downstream stations might thus be larger than indicated by the model.

257 This paper has → We have

263 It was shown → We show

271 How relevant are (seasonal) changes of roughness and bedforms, due to changes in water and sediment supply by the dam?

**Figure 2** It would be more meaningful to plot ($z_{\mathrm{pred}} - z_{\mathrm{obs}}$) vs $z_{\mathrm{obs}}$ and to use smaller dots which do not overlap that much. This would reveal better any systematic variation.

**Figure 3** Add subplots titles, like Discharge, Downstream level, Upstream level so that the figure can be interpreted without looking up the meaning of the coefficients $\alpha, \beta, \gamma$.

**Figure 3** begins from Jiangyin $\rightarrow$ upstream of Jiangyin

**Figure 7** The average annual average hydrograph of the post-TGD period is corrupted by high-frequent fluctuations of the hydrograph. The graph would be clearer if the fluctuation is removed it through by smoothing with a sliding window. A triangular window with a width of 30 days seems appropriate. Smooth the data for the pre-TGD period as well, for better comparison.

**References**

Cai, H., H. H. G. Savenije, C. Jiang, L. Zhao, and Q. Yang, Analytical approach for determining the mean water level profile in an estuary with substantial fresh water discharge, *Hydrology and Earth System Sciences*, *20*(3), 1177–1195, 2016.

Efron, B., and R. J. Tibshirani, *An Introduction to the Bootstrap*, Chapman & Hall/CRC Monographs on Statistics & Applied Probability, Taylor & Francis, 1994.

Kästner, K., A. J. F. Hoitink, P. J. J. F. Torfs, E. Deleersnijder, and N. S. Ningsih, Propagation of tides along a river with a sloping bed, *Journal of Fluid Mechanics*, *872*, 39–73, 2019.

Kukulka, T., and D. A. Jay, Impacts of Columbia River discharge on salmonid habitat: 2. Changes in shallow-water habitat, *Journal of Geophysical Research: Oceans*, *108*(C9), 1–17, 2003.

LeBlond, P. H., On tidal propagation in shallow rivers, *Journal of Geophysical Research: Oceans*, *83*(C9), 4717–4721, 1978.

Savenije, H. H. G., *Salinity and Tides in Alluvial Estuaries, 2nd completely revised edition*, salinityandtides.com, 2012.

---

## Author Comment (AC1)

**Response letter to Reviewer#1**

We thank Reviewer#1 for the careful consideration of our work. We agree with his/her constructive and thoughtful comments and suggestions, which led to a much improved and complete manuscript. In this response letter, we have replied (in blue) to all the comments formulated by the Reviewer (in black).

**Comments:**

Thank you for sending me the manuscript: "Quantifying the impacts of the Three Gorges Dam on the spatial-temporal water level dynamics in the Yangtze River estuary" by Huayang Cai for review, which I read with great interest.

The authors apply a linear regression model to the tidally averaged water level in the Yangtze estuary to investigate the effects of the Three Gorges dam. The authors find that their regression model predicts the water level in the Yangtze reasonably well. They find that since construction of the dam, low flows have increased while flows during transition from the high to the low flow season have decreased.

The topic is very relevant and the manuscript was interesting to read. The applicability of a regression model to predict water levels in tidal rivers agrees with my own experience in this field. The text and figures are of high quality.

However, the regression model applied here is relatively simple, at least much simpler than previously applied models. This certainly makes it easy to grasp the results, especially for readers who are not experts on the topic. However, this also makes it difficult to identify the physical drivers behind changes in the water levels, and might introduce systematic errors. Below, I provide suggestions on how these issues can be verified and mitigated, if necessary.

Our reply: We very much appreciate all the comments and suggestions raised by the reviewer. In the revised manuscript, we shall completely address all the comments.

**Methods**

1. The regression model includes both discharge and water level at the upstream station. As they depend on each other, the model is not parsimonious. As a consequence, the columns for $Q$ and $Z_{up}$ of the regression matrix will be close to collinear so that small changes (errors) in the data can result in large changes in the coefficients $\alpha$ and $\gamma$ even if the fit is good. Possible changes of the coefficients over time might thus be regression

artefacts. This should be ruled out by verifying that $Q_{up}$ and $Z_{up}$ are not strongly correlated.

- ■ If the correlation is weak, then the model is robust, but then it would be insightful to elaborate on why the upstream water level and discharge are unrelated. The comment "influenced by the dynamics of [] tributaries" (l. 119) is unclear. The water level is uniquely determined by the backwater curve as long as the daily averaged water level does not change rapidly in time. Therefore, tributaries upstream of the inflow boundary influence downstream levels only through their discharge. Do the authors refer to tributaries downstream of the upstream station?

Our reply: It can be seen from Figure R1 below that the correlation between $Q_{up}$ and $Z_{up}$ is indeed strong. However, we observe that the daily averaged water levels are not uniform for identical river discharge (see Figure R1a) due to the external forcing, either the potential influence induced by the tidal forcing or the exerted residual water level slope upstream of the DT hydrological station. Actually, the observed water levels at DT hydrological stations were influenced by both the residual water level slope upstream of the inflow boundary (owing to the relative importance of river discharge between the main stream and the tributaries, especially during the flood season) and that downstream of the inflow boundary (owing to the tidal forcing, especially during the dry season). To account for the influence of residual water level slope, in the previous manuscript we have explicitly introduced the $z_{up}$ into the regression model.

[Figure]

Figure R1. Relationship between water level and river discharge at the DT hydrological station (a) and that between residual water level slope for the whole estuary and river discharge (b).

■ If the correlation is strong, then it is better to replace the terms $\alpha Q + \gamma Z_{up}$ with the non-linear term $aQ^b$. This model is less ambiguous. In my personal experience, the coefficients $a$ and $b$ of the non-linear model also give much more insight into the influence of the river discharge on the mean water level along tidal rivers.

Our reply: We very much appreciate the comments raised by the Reviewer. In this case, the regression model can be described by the following equation:

$$Z = Z_0 + \alpha Q^\beta + \gamma Z_{down} \tag{R1}$$

where the potential influence of $Z_{up}$ on water level dynamics is implicitly accounted by the nonlinear term $\alpha Q^\beta$. It can be seen from Figure R2 and Table R1 that the model performance is more or less the same as the original triple linear regression model, except that the RMSE values are slightly larger at NJ, MAS and WH stations (ranging between 0.17 and 0.21 m) than those using the triple linear regression model (ranging between 0.11 and 0.15 m).

[Figure]

Figure R2. Comparison between predicted and observed daily averaged water levels for both the pre-TGD and post-TGD periods at different gauging stations along the YRE by means of Equation R1: (a) Jiangyin (JY), (b) Zhenjiang (ZJ), (c) Nanjing (NJ), (d) Maanshan (MAS), (e) Wuhu (WH).

Table R1. Calibrated regression coefficients for both the pre-TGD and post-TGD periods along the YRE by means of Equation R1.

| Stations | | $Z_0$ | $\alpha$ | $\beta$ | $\gamma$ | RMSE/m | Standard deviation/m |
|---|---|---|---|---|---|---|---|
| JY | pre-TGD | -0.10 | 0.31 | 0.62 | 0.49 | 0.06 | 0.64 |
| | post-TGD | -0.44 | 0.55 | 0.45 | 0.43 | 0.08 | 0.59 |
| ZJ | pre-TGD | 0.00 | 0.89 | 0.92 | 0.48 | 0.13 | 1.23 |
| | post-TGD | -0.27 | 1.02 | 0.82 | 0.43 | 0.14 | 1.12 |
| NJ | pre-TGD | -0.37 | 1.84 | 0.81 | 0.40 | 0.17 | 1.72 |
| | post-TGD | -0.55 | 1.57 | 0.84 | 0.40 | 0.19 | 1.54 |
| MAS | pre-TGD | -0.57 | 2.50 | 0.77 | 0.38 | 0.20 | 2.02 |
| | post-TGD | -0.64 | 2.03 | 0.83 | 0.37 | 0.21 | 1.80 |
| WH | pre-TGD | -1.31 | 3.78 | 0.66 | 0.32 | 0.21 | 2.35 |
| | post-TGD | -1.36 | 3.09 | 0.71 | 0.32 | 0.21 | 2.07 |

2. The regression model does not include the effect of the tides on the mean water level. However, this effect is not negligible during periods of low river flow (LeBlond, 1978). This introduces a systematic error. As the Three Gorges dam increased river discharge during the low flow season, this can bias the results. It is, therefore, reasonable to include the influence of tides on the mean water level in the regression model. For example, Kukulka and Jay (2003) suggest the regression model inear in $h^3$:

$$h^3 \approx aQ_{river}^2 + b\,|\,z_{tide}\,|^2 + c,$$

while (Kastner et al., 2019) suggested linearizing the backwater equation, which can be readily approximated in a regression model linear in $h$ (or $z$).

Our reply: Actually, the potential effect of the tides on the mean water level is implicitly considered by the $Z_{down}$ term, which is typically featured by a spring-neap cycle. Figure R3 shows the autoregressive power spectral density estimate of the daily averaged water level observed at TSG gauging station, where significant periodic cycle of 14.8 days was observed.

[Figure]

Figure R3. Autoregressive power spectral density estimate of the daily averaged water level observed at TSG gauging station

3. The independent variables are not normalized in the regression model so that the coefficients have very different magnitudes ($O(\alpha) = 10^{-5}$ while $O(\beta) = 1$). It is thus not obvious which predictor (downstream or upstream level) has the largest influence at a particular location. This can be revealed by normalizing the independent variables by their standard deviation before the regression:

$$Z = Z_0 + \alpha Q / \mathrm{std}(Q) + \beta Z_{\mathrm{down}} / \mathrm{std}(Z_{\mathrm{down}}) + \gamma Z_{\mathrm{up}} / \mathrm{std}(Z_{\mathrm{up}})$$

This is preferable to the order in the study, where variance is normalized after the regression.

Our reply: We thank the reviewer to point this out. In the revised manuscript, we shall normalize the input parameters by their standard deviations.

4. Interpolation of slopes and uncertainty estimates (Figure 4 and 5, lines 190_)
- ■ There is a mistake in the slope calculation. The values should be in the order $10^{-5}$, not $10^{-8}$. The distance between the stations was probably not converted from km to m.

Our reply: You are right! In the revised manuscript, we shall correct this mistake (see Figure R4 below).

[Figure]

Figure R4. Reconstructed spatial-temporal water levels, Z, (a, c) and their slopes, S, (b, d) for the climatological year during both the pre-TGD (a, b) and post-TGD (c, d) periods. The red lines in subplots (b) and (d) indicate the local minimum water level slopes in the central section of the YRE (between Jiangyin and Zhenjiang).

■ Determining the slope from by higher-order (Hermite) interpolation is not meaningful here. This is because the error (of the slope) is amplified at the interpolated values between the stations. As a consequence, the interpolated slope has unrealistic local extrema at the midpoints between stations (Figure 5). The error of the cubically (Hermite) interpolated slope at the midpoint between stations is about 1.8 times as large as the errors of the levels at the stations. Since the error of the levels is about 10%, the error of the slope is about 20%. The local maxima of the slope, as well as the difference between the pre- and post-TGD period as indicated in Figure 5 are therefore insignificant. The interpolation error (of the slopes) can be considerably reduced by calculating the slopes at the midpoint between two stations and then linearly interpolating the slopes between the midpoints. In this case the error of the slope is only 0.7 times that of the error in levels. If the authors want to retain cubic interpolation, then the spurious extrema can be suppressed by fitting the 4 coefficients of the cubic polynomial with all 5 stations in a least squares manner.

Our reply: Many thanks for the reviewer's comments on the interpolation of the results. Actually, we only interpolated the daily averaged water level along the estuary, while the slope is computed on the basis of the interpolated water level using the Matlab

"gradient.m" function. However, it is true that the unrealistic local extrema are mainly due to the amplification of the error of the interpolated water level.

■ As the model is not parsimonious, it might fit well even if the regression coefficient are uncertain, as multiple parameter combinations can result in similar good model performance. A good way to assess the uncertainty is bootstrapping (Efron and Tibshirani, 1994). Simply split the time series into blocks comprising of one month, this reduces the effect of serial correlation. When there are n blocks, randomly choose $\sqrt{n}$ blocks and fit the model. Repeat this a few hundred times. The standard error is simply the standard deviation of the estimated parameters. The standard error of the coefficient, predicted levels and slopes can then be indicated with error bars in Figure 3 and 5. The cubic interpolation results in larger errors at midpoints between sections, so errors bars are best placed there.

Our reply: We very much appreciate the reviewer's comments on the quantification of the model uncertainty. Actually, since we used the default multiple linear regression function "regress.m" in Matlab, it is possible to output a matrix of 95% confidence intervals for the coefficient estimates, which is similar to the adoption of bootstrapping. In the revised manuscript, we shall update the Figure 3 by including the error bar for each coefficient (see Figure R5 below).

[Figure]

Figure R5. Interpolated linear regression coefficients $Z_0$ (a), $\alpha$ (b), $\beta$ (c), $\gamma$ (d) with error bar along the YRE (upstream of the Jiangyin gauging station) for both the pre-TGD and post-TGD periods. The vertical error bar was estimated using the Matlab 'regress.m' function with 95% confidence intervals.

**Minor**

■ Title estuary➜upper estuary

Our reply: We agree with the reviewer's comment. The title will be revised as: "Quantifying the impacts of the Three Gorges Dam on the spatial-temporal water level dynamics in the upper Yangtze River estuary"

■ 35 The term "analytical solution" is misleading, as the water level is still determined by (numerically) integrating an initial value problem (eq. 22 in Cai et al. (2016)).

Our reply: In the revised manuscript, we shall replace "analytical solutions" with "solutions".

■ 44 Kukulka and Jay (2003) should be referenced here, as an important regression model for the mean water level of tidal rivers.

Our reply: In the revised manuscript, we shall include the reference of Kukulka and Jay (2003).

■ 45 "these methods suggest that water level dynamics in estuaries are highly nonlinear and nonstationary" This sounds as if water levels in tidal are difficult to analyse and predict, and that looking at tidal cycle/average is a novel idea. However, there is a large amount of publications how water levels can be approximated well on a cycle-by-cycle basis, see the works of the groups of Savenije, Godin, Jay, Hoitink, and Friedrichs.

Our reply: In the revised manuscript, we shall revise this sentence as: "these methods suggest that water level dynamics in estuaries are highly nonlinear and nonstationary owing to complex tide-river interactions".

■ 49 The reference to Darcy is dubious. Even if the surface level can be predicted by a linear regression model, it is still turbulent flow (quadratic flow resistance), which is very different from groundwater ow (linear flow resistance).

Our reply: In the revised manuscript, we shall remove "similar to Darcy's law for groundwater flow".

■ 89 Mark Gaoqiaoju on the map in Figure 1

Our reply: In the revised manuscript, we shall mark Gaoqiaoju gauging station on the Map.

■ 93 "we mainly concentrate on the tide-river dynamics" This is not the case, since, as commented by me before, the tidally induced water level offset is not included in the regression model.

Our reply: In the revised manuscript, to be more specific, we shall replace "tide-river dynamics" with "water level dynamics". Actually, since the input parameter $Z_{down}$ implicitly considered the influence induced by the tidal forcing (especially the spring-neap changes), we actually concentrated on the tide-river dynamics.

■ 110 Mention here, which of the stations where chosen as the upstream and the downstream end (Datong and Gaoqiaoju?).

Our reply: We agree with the reviewer's comment. In the revised manuscript, we shall explicitly mention that "In this study, the DT hydrological station was chosen as the upstream end, while the TSG gauging station being the downstream end."

■ 169 "linear" is misleading here. The water depth in the upstream estuary most likely scale like $h \approx Q^{2/3}$. The non-linearity is just hidden by including $Z_{up}$ in the regression model.

Our reply: In the revised manuscript, we shall explicitly mention that "which leads support to our hypothesis that the response of water level dynamics to hydrodynamics at both ends of the estuary is largely linear in the YRE owing to the explicit inclusion of $Z_{up}$ in the regression model."

■ 248 The conclusion "[at the downstream stations] tide dominates [the tidally averaged water level]" sounds odd, as the regression model applied in this study does not explicitly account for the tidally induced water level offset. It only includes the tidally averaged water level at the seaward station. However, at the river mouth the tidally induced water level offset is negligible as it integrates along the estuary, c.f. Kastner et al. (2019) and Cai et al. (2016). So, no meaningful conclusion about the tidal influence can be drawn. The model probably indicates that fluctuations of the sea level unrelated to tides, such as wind, ocean-temperature and ocean-salinity, dominate the mean water level dynamics near the sea. It would be insightful to actually determine the tidal influence by including it explicitly in the regression model.

Our reply: Actually, since the input parameter $Z_{down}$ in the regression model implicitly considered the influence induced by the tidal forcing (especially the spring-neap changes), we actually concentrated on the tide-river dynamics.

■ 248 The river discharge influences the salinity gradient, and with it the variation of the water level at the reference station at the sea (Savenije, 2012). The influence on river discharge on the downstream stations might thus be larger than indicated by the model.

Our reply: We agree with the reviewer that the salinity gradient may influence the water level at the reference station at the sea. However, since the study area is out of the maximum salt intrusion length, thus the potential influence due to salinity gradient is negligible.

■ 257 This paper has →We have

Our reply: In the revised manuscript, we shall replace "This paper has" with "In this study, we have".

■ 263 It was shown →We show

Our reply: We agree with the reviewer's comment.

■ 271 How relevant are (seasonal) changes of roughness and bedforms, due to changes in water and sediment supply by the dam?

Our reply: Here we can conclude that the main impact due to changes in water and sediment supply by the dam tends to deepen the riverbed since the alterations caused by geometric changes are negative.

■ Figure 2 It would be more meaningful to plot ($z_{pred}$ - $z_{obs}$) vs $z_{obs}$ and to use smaller dots which do not overlap that much. This would reveal better any systematic variation.

Our reply: We agree with the reviewer's comments. In the revised manuscript, the Figure 2 will be revised as follows (see Figure R6 below).

[Figure]

Figure R6. Alterations in difference between predicted and observed daily averaged water levels as a function of observed daily averaged water levels for both the pre-TGD and post-TGD periods at different gauging stations along the YRE: (a) Jiangyin (JY), (b) Zhenjiang (ZJ), (c) Nanjing (NJ), (d) Maanshan (MAS), (e) Wuhu (WH).

■ Figure 3 Add subplots titles, like Discharge, Downstream level, Upstream level so that the figure can be interpreted without looking up the meaning of the coefficients $\alpha$, $\beta$, $\gamma$.

Our reply: We agree with the reviewer's comment. In the revised manuscript, we shall include the subplots titles (see Figure R5 above).

■ Figure 3 begins from Jiangyin➔upstream of Jiangyin

Our reply: We agree with the reviewer's comment.

■ Figure 7 The average annual average hydrograph of the post-TGD period is corrupted by high-frequent fluctuations of the hydrograph. The graph would be clearer if the fluctuation is removed it through by smoothing with a sliding window. A triangular window with a width of 30 days seems appropriate. Smooth the data for the pre-TGD period as well, for better comparison.

Our reply: We agree with the reviewer's comments. In the revised manuscript, the Figure 7 will be revised as follows (see Figure R7 below).

[Figure]

Figure R7. Alterations in river discharge and water level observed at DT and TSG, respectively, during the post-TGD period relative to the pre-TGD period over the climatological year. The daily averaged river discharge and water level were smoothed using a moving average filter with a span of 30 days.

References:
Cai, H., Savenije, H. H. G., Jiang, C., Zhao, L., and Yang, Q.: Analytical approach for determining the mean water level profile in an estuary with substantial fresh water discharge, Hydrol. Earth Syst. Sci., 20, 1177-1195, https://doi.org/10.5194/hess-20-1177-2016, 2016.
Kastner, K., Hoitink, A.J.F., Torfs, P.J.J.F., Deleersnijder, E., Ningsih, N.S.: Propagation of tides along a river with a sloping bed. J. Fluid Mech., 872, 39-73, https://doi.org/10.1017/jfm.2019.331, 2019.

Kukulka, T., and Jay, D. A.: Impacts of Columbia River discharge on salmonid habitat: 2. Changes in shallow-water habitat, J. Geophys. Res., 108(C9), 3294, https://doi.org/10.1029/2003JC001829, 2003.

Savenije, H.H.G., 2012. Salinity and tides in alluvial estuaries. completely revised 2nd edition. Available from http://www.salinityandtides.com [Accessed 11 June 2022].

---

## Author Comment (AC2)

**Response letter to Reviewer#2**

We thank Reviewer#2 for the careful consideration of our work. We agree with her constructive and thoughtful comments and suggestions, which led to a much improved and complete manuscript. In this response letter, we have replied (in blue) to all the comments formulated by the Reviewer (in black).

**Comments:**
In this study, the authors investigated the spatial-temporal water level dynamics along the main stream of the Yangtze River estuary by means of a triple linear regression model accounting for both the upstream and downstream boundary conditions. The model was subsequently used to quantify the influence of the Three Gorge Dam's operation on the water level dynamics. Results showed that the alteration in water level dynamics are mainly controlled by the variation in freshwater discharge owing to the Three Gorge Dam's operation, while the influence by geometric changes are minor when compared with that of the river discharge alteration. The first reviewer already provided many constructive comments on the manuscript, which I mostly agreed, especially concerning the validity of the proposed triple linear regression model. Generally, the paper is well organized and written. However, there are still some concerns which should be properly addressed before the paper can be accepted in the Ocean Science.
Our reply: We very much appreciate all the comments and suggestions raised by the reviewer. In the revised manuscript, we shall completely address all the comments.

**Major concerns:**
1. The authors assumed that the alteration in water level dynamics can be primarily attributed to the geometric change (caused by the combined influences of both natural and anthropogenic modifications) and the boundary effects (induced by the changes in upstream and downstream conditions, primarily due to the TGD's freshwater regulation). Since the authors proposed a triple linear regression model to quantify the impacts of the Three Gorges Dam (representing the intensive human intervention) on the water level dynamics, how did the authors account for the potential impacts due to the climate change (such as intensifying precipitation, global sea level rise, etc.)?
Our reply: We thank the reviewer for pointing this out. Indeed, for the time being, we assumed that the largest contribution to the alteration of river discharge before and after the TGD can be primarily attributed to the TGD's freshwater regulation, which is not completely true due to the influences of other dams (such as Gezhouba dam) and the climate change (such as intensifying precipitation over the river basin). Similarly, the potential influence of climate change (such as global sea level rise) may slightly alter the water level at the downstream boundary. Consequently, in the revised manuscript, we shall clarify that: "*It is worth noting that the quantity $\Delta_{BOU}$ (including both the upstream and downstream boundary conditions) should be interpreted as the water*

*level alteration owing to the overall influences driven by both human interventions and climate change. However, in this study the largest contribution to the alteration in upstream boundary condition (i.e., river discharge) can be primarily attributed to the TGD's operation, since the TGD alone accounts for more than 30% of the total storage capacity of the dams constructed between 1987 and 2014 along the Yangtze River (Li et al., 2016). In addition, we note that the only other dam (Gezhouba, abbreviated by GZB, see Figure 1a) along the main course of the Yangtze River was constructed in 1981 (before the TGD). With regard to the downstream boundary condition, the adopted water levels observed at TSG station implicitly account for the potential impacts induced by both anthropogenic (such as channel dredging) and climate (such as global sea level rise) changes.*"

2. It was mentioned by the authors that the proposed model is particularly useful for determining scientific strategies for sustainable water resources management in dam-controlled estuaries worldwide. Actually, as far as I see, the proposed method can also be used to quantify the influence of climate change on spatial-temporal water level dynamics since both the upstream and downstream boundary conditions are closely related to the climate change even without the construction of large dams. Further comments with regard to the applicability of the proposed method can be clarified.

Our reply: We agree with the reviewer's comment. In the revised manuscript, we shall clarify that: "*Such a novel approach should be particularly helpful for determining scientific guidelines for sustainable water resources management (e.g., dredging for navigation, flood control, salt intrusion prevention etc.) in estuaries worldwide, especially for dam-controlled estuaries. In addition, the proposed method can also be used to quantify the potential impacts of changes in boundary conditions induced by climate change (such as intensifying precipitation, global sea level rise, etc.) in natural estuaries without considerable human interventions*".

In addition, we shall slightly modify the last sentence in the abstract part: "*The presented method to quantify the separate contributions made by changes in boundary conditions and geometry on spatial-temporal water level dynamics is particularly useful for determining scientific strategies for sustainable water resources management in dam-controlled or climate-driven estuaries worldwide*".

3. The geometric effect in this paper is mainly referred to the bathymetric changes in the estuarine system, which should be the primary factor dominating the geomorphological changes in the Yangtze river estuary. However, for other estuarine systems, the geometric effect could also due to the lateral boundary changes. Could the authors give some comments on the applicability of the proposed method to such cases?

Our reply: In the revised manuscript, we shall clarify that: "*Meanwhile, it is also worth noting that the quantity $\Delta_{GEO}$ should be interpreted as the water level alteration due to the overall impacts caused by both the bathymetric change and the storage area change.*"

4. Finally, I would suggest the authors to clarify the implications of this contribution.

Our reply: We very much appreciate this suggestion raised by the reviewer. In the revised manuscript, we shall explicitly mention that: " *There exists a long tradition of statistical, analytical and numerical studies on tide-river interactions in estuaries worldwide, such as the Columbia River estuary in the USA (e.g., Kukulka and Jay, 2003; Jay et al., 2015; Pan et al., 2018b), the St. Lawrence River estuary in Canada (e.g., Godin,1999; Matte et al., 2013, 2014), the Mahakam River estuary in Indonesia (e.g., Buschman et al., 2009; Sassi and Hoitink, 2013), the Yangtze River estuary in eastern China (e.g., Guo et al., 2015, 2020; Yu et al., 2020) and the Pearl River estuary in southern China (e.g., Zhang et al., 2018; Cai et al., 2018b, 2019b). These studies showed that as tides propagate along the estuary the tidal amplitude, phase and shape were influenced by the bottom friction, channel geometry and river discharge. In this study, with the proposed simple yet effective triple linear regression model, we are able to isolate and to quantify the impacts of the boundary (such as freshwater regulation due to dam's operation) and geometric (such as channel dredging) effects on the tide-river dynamics. Such a novel approach should be particularly helpful for determining scientific guidelines for sustainable water resources management (e.g., dredging for navigation, flood control, salt intrusion prevention etc.) in estuaries worldwide, especially for dam-controlled estuaries. In addition, the proposed method can also be used to quantify the potential impacts of changes in boundary conditions induced by climate change (such as intensifying precipitation, global sea level rise, etc.) in natural estuaries without considerable human interventions.* ".

References:

Buschman, J. F., van der Vegt, M., and Hoekstra, P.: Subtidal water level variation controlled by river flow and tides, Water Resour. Res., 45, W10 420, https://doi.org/Artn W1042010.1029/2009wr008167, 2009.

Cai, H., Yang, Q., Zhang, Z., Guo, X., Liu, F., Ou, S.: Impact of river-tide dynamics on the temporal-spatial distribution of residual water levels in the Pearl River channel networks, Estuar. Coasts, 41, 1885-1903, https://doi.org/10.1007/s12237-018-0399-2, 2018.

Cai, H., Yang, H., Liu, J., Niu, L., Ren, L., Liu, F., Ou, S., Yang, Q.: Quantifying the impacts of human interventions on relative mean sea level change in the Pearl River Delta, China, Ocean Coast. Manage., 173, 52-64, https://doi.org/10.1016/j.ocecoaman.2019.02.007, 2019.

Godin, G.: The propagation of tides up rivers with special considerations on the upper

Saint Lawrence river, Estuar. Coast. Shelf S., 48, 307 – 324.

https://doi.org/10.1006/ecss.1998.0422, 1999.

Guo, L.C., van der Wegen, M., Jay, D.A., Matte, P., Wang, Z.B., Roelvink, D., He, Q.: River-tide dynamics: Exploration of nonstationary and nonlinear tidal behavior in the

Yangtze River estuary, J. Geophys. Res. -Oceans, 120, 3499 – 3521.

https://doi.org/10.1002/2014jc010491, 2015.

Guo, L. C., Zhu, C. Y., Wu, X. F., Wan, Y. Y., Jay, D. A., Townend, I., Wang, Z. B., and He, Q.: Strong inland propagation of low-frequency long waves in river estuaries, Geophys. Res. Lett., 47, e2020GL089112, https://doi.org/10.1029/2020GL089112, 2020.

Jay, D.A., Leffler, K., Diefenderfer, H.L., Borde, A.B.: Tidal-fluvial and estuarine processes in the lower Columbia River: I. along-channel water level variations, Pacific Ocean to Bonneville Dam, Estuar. Coast., 38, 415-433, https://doi.org/10.1007/s12237-014-9819-0, 2015.

Li, Z., Zhang, Z. Y., Lin, C. X., Chen, Y. B., Wen, A. B. and Fang, F.: Soil-air greenhouse gas fluxes influenced by farming practices in reservoir drawdown area: a case at the three Gorges Reservoir in China, J. Environ. Manage., 181, 64–73, https://doi.org/10.1016/j.jenvman.2016.05.080, 2016.

Kukulka, T., and D. A. Jay, Impacts of Columbia River discharge on salmonid habitat: I. A nonstationary fluvial tide model, J. Geophys. Res., 108(C9), 3293, https://doi.org/10.1029/2002JC001382, 2003.

Matte, P., Jay, D. A., and Zaron, E. D.: Adaptation of classical tidal harmonic analysis to nonstationary tides, with application to river tides, J. Atmos. Ocean. Tech., 30, 569–589, https://doi.org/10.1175/Jtech-D-12-00016.1, 2013.

Matte, P., Secretan, Y., and Morin, J.: Temporal and spatial variability of tidal-fluvial dynamics in the St. Lawrence fluvial estuary: An application of nonstationary tidal harmonic analysis, J. Geophys. Res.-Oceans, 119, 5724–5744, https://doi.org/10.1002/2014jc009791, 2014.

Pan, H. D., Lv, X. Q., Wang, Y. Y., Matte, P., Chen, H. B., and Jin, G. Z.: Exploration of tidal-fluvial interaction in the Columbia River estuary using S TIDE, J. Geophys. Res.-Oceans, 123, 6598–6619, https://doi.org/10.1029/2018jc014146, 2018b.

Sassi, M. G. and Hoitink, A. J. F.: River flow controls on tides and tide-mean water level profiles in a tidal freshwater river, J. Geophys. Res.-Oceans, 118, 4139–4151, https://doi.org/10.1002/jgrc.20297, 2013.

Yu, X., Zhang, W., Hoitink, A.J.F.: Impact of river discharge seasonality change on tidal duration asymmetry in the Yangtze river estuary, Sci. Rep., 10, 6304, https://doi.org/10.1038/s41598-020-62432-x, 2020.

Zhang, W., Cao, Y., Zhu, Y.L., Zheng, J.H., Ji, X.M., Xu, Y.W., Wu, Y., Hoitink, A.J.F.: Unravelling the causes of tidal asymmetry in deltas, J. Hydrol., 564, 588 – 604. https://doi.org/10.1016/j.jhydrol.2018.07.023, 2018.

---

## Referee Report (RR1)

Dear Editor Prof. Huthnance,

Thank you for sending me the revised manuscript "Quantifying the impacts of the Three Gorges Dam on the spatial-temporal water level dynamics in the upper Yangtze River estuary" by Huayang Cai et al. for review.

The authors investigate changes in the water level dynamics in the upper Yangze estuary by means of a linear regression model due to construction of the TGD. They separate the changes into geometrical induced by scouring and hydrodynamically induced due to discharge modulation by the TGD. They show that on average, water levels in the TGD dropped.

The results seem plausible and I like the idea to separate the effects. The authors also responded to my comments to the first version in detail and improved their manuscript accordingly. Therefore, I recommend to publish the manuscript with minor clarifications. My comments can be addressed without another review round.

Kind regards,

Reviewer

**Comments**

95 2.2 Dataset In the study both discharge and water level at the upstream station is used, which implies that there is are continuous measurements of both the stage and flow velocity. Yet measurements methods can change over time, and even if they do not change, they require frequent recalibration to account for morphological changes at the gauging station. Therefore, it would be insightful to provide some information on how discharge at the Datong station is measured, if the method of measurement changed during the study period, and most importantly, if it was regularly updated to account for scouring of the bed, after the TGD had been constructed.

120 "daily averaged water levels observed at the DT hydrological station are not uniform for identical river discharge"
→ There is no unique stage-discharge relation at the Datong hydrological station

120 "due to the influence of external forcing [...]"
A potentially important factor, the stage-discharge hysteresis, is not mentioned. Is it not relevant at Datong? I suggest to provide a rough estimate of the stage-discharge hysteresis.

116 standard deviation function → standard deviation

139 variance function → variance

176 Note that Gezhouba is also a run-of-the-river dam, and therefore should not considerably influence the discharge regime.

212 increased → increasing

215 The standard error [...] represents the standard deviation
→ The error-bars [...] represent the standard error
The standard error and standard deviation are related but not the identical ($s_{err} \propto s_d/\sqrt{n_{sample}}$).

215 is robust → is fitting well
"Robust" in statistics implies that a method to suppress outliers was employed, which is not the case here.

215 "[...] the standard error [...] suggests that the proposed triple linear regression model is [fitting well] with limited uncertainty"
Remove the qualifier "with limited uncertainty", as as a good fit does not imply low uncertainty. In general, the goodness of fit to the measured values improves when more more parameters are added, but the reliability of predicting values at moments for which no measurements are available decreases (overfit). (See also my recommendation in the previous revision to validate the model through bootstrapping.)

310 constant value of local mean sea level → constant mean sea level

310 "[The] channel deepening [...] tend[s] to increase in the landward direction [..]. This phenomenon can be primarily attributed to the constant value of local mean sea level or the ultimate base level that the topography tends to approach due to erosion."
I cannot follow this argument, as the constant sea level in combination with the seasonal discharge variation promotes, not prevents, scouring c.f. theoretical work by (*Lamb et al.*, 2012) for the Mississippi and measured longitudinal river profiles of the Mahakam (*Sassi et al.*, 2012) and Kapuas (*Kästner et al.*, 2017). I propose two alternative hypotheses: First, reduced sediment supply initially just results in scouring downstream in the vicinity of the dam, after which the scour slowly propagates further downstream with time. Second, the reduction of seasonal discharge variation by the TGD reduces the overdeepening near the sea.

294 Conclusion: Since the TGD continues to deprive the Yangtze of sediment, it is reasonable to assume that the scouring will continue. Can the authors hypothesize how the water levels will evolve in future?

This also points to a potential methodological limitation of the study, as the mean conditions are treated as if they were stationary before and after the dam construction, while the geometric influence has likely gradually increased since construction of the dam due to ongoing scouring.

**References**

Kästner, K., A. J. F. Hoitink, B. Vermeulen, T. J. Geertsema, and N. S. Ningsih, Distributary channels in the fluvial to tidal transition zone, *Journal of Geophysical Research: Earth Surface*, *3*(122), 696–710, 2017.

Lamb, M. P., J. A. Nittrouer, D. Mohrig, and J. Shaw, Backwater and river plume controls on scour upstream of river mouths: Implications for fluviodeltaic morphodynamics, *Journal of Geophysical Research: Earth Surface (2003–2012)*, *117*(F1), 2012.

Sassi, M. G., A. J. F. Hoitink, B. Brye, and E. Deleersnijder, Downstream hydraulic geometry of a tidally influenced river delta, *Journal of Geophysical Research: Earth Surface*, *117*(F4), 2012.

---

## Author Response (AR2)

**Response letter**

We thank the Editor and the Reviewers for the careful consideration of our work. Their constructive and thoughtful comments and suggestions led to a much improved and complete revision of the manuscript. In the revised paper, we have addressed all the comments formulated by the Reviewers by replying (in black) to their remarks (in blue). The lines numbers in this rebuttal refer to the revised version of the manuscript.

**Response letter to Reviewer#1**

**Comments:**

Thank you for sending me the manuscript: "Quantifying the impacts of the Three Gorges Dam on the spatial-temporal water level dynamics in the Yangtze River estuary" by Huayang Cai for review, which I read with great interest.

The authors apply a linear regression model to the tidally averaged water level in the Yangtze estuary to investigate the effects of the Three Gorges dam. The authors find that their regression model predicts the water level in the Yangtze reasonably well. They find that since construction of the dam, low flows have increased while flows during transition from the high to the low flow season have decreased.

The topic is very relevant and the manuscript was interesting to read. The applicability of a regression model to predict water levels in tidal rivers agrees with my own experience in this field. The text and figures are of high quality.

However, the regression model applied here is relatively simple, at least much simpler than previously applied models. This certainly makes it easy to grasp the results, especially for readers who are not experts on the topic. However, this also makes it difficult to identify the physical drivers behind changes in the water levels, and might introduce systematic errors. Below, I provide suggestions on how these issues can be verified and mitigated, if necessary.

Our reply: We very much appreciate all the comments and suggestions raised by the reviewer. In the revised manuscript, we have completely addressed all the comments.

**Methods**

1. The regression model includes both discharge and water level at the upstream station. As they depend on each other, the model is not parsimonious. As a consequence, the columns for $Q$ and $Z_{up}$ of the regression matrix will be close to collinear so that small changes (errors) in the data can result in large changes in the coefficients α and γ even if the fit is good. Possible changes of the coefficients over time might thus be regression artefacts. This should be ruled out by verifying that $Q_{up}$ and $Z_{up}$ are not strongly correlated.

- ■ If the correlation is weak, then the model is robust, but then it would be insightful to elaborate on why the upstream water level and discharge are unrelated. The

comment "influenced by the dynamics of [] tributaries" (l. 119) is unclear. The water level is uniquely determined by the backwater curve as long as the daily averaged water level does not change rapidly in time. Therefore, tributaries upstream of the inflow boundary influence downstream levels only through their discharge. Do the authors refer to tributaries downstream of the upstream station?

Our reply: It can be seen from Figure R1 below that the correlation between $Q_{up}$ and $Z_{up}$ is indeed strong. However, we observe that the daily averaged water levels are not uniform for identical river discharge (see Figure R1a) due to the external forcing, either the potential influence induced by the tidal forcing or the exerted residual water level slope upstream of the DT hydrological station. Actually, the observed water levels at DT hydrological stations were influenced by both the residual water level slope upstream of the inflow boundary (owing to the relative importance of river discharge between the main stream and the tributaries, especially during the flood season) and that downstream of the inflow boundary (owing to the tidal forcing, especially during the dry season). To account for the influence of residual water level slope, in the previous manuscript we have explicitly introduced the $z_{up}$ into the regression model.

In the revised manuscript, we have explicitly mentioned that: "*The source code of the proposed triple linear regression model is available at https://github.com/Huayangcai/Triple-Linear-Regression-Model-V1.0-Matlab-Toolbox. It is worth noting that daily averaged water levels observed at the DT hydrological station are not uniform for identical river discharge (see Figure S1 in the Supplementary Material) due to the influence of external forcing, either the potential influence induced by the tidal forcing (especially during the dry season) or the exerted residual water level slope upstream of the DT hydrological station (owing to the relative importance of river discharge between the main stream and the tributaries, especially during the flood season). Thus, in order to explicitly account for the influence of extern forcing in both upstream and downstream reaches, here we have explicitly introduced the $z_{up}$ into the regression model, and hence the dynamics of residual water level slope along the upper YRE.*" (see Lines 118-127)

[Figure]

Figure R1. Relationship between water level and river discharge at the DT hydrological station (a) and that between residual water level slope for the whole estuary and river discharge (b).

- If the correlation is strong, then it is better to replace the terms $\alpha Q + \gamma Z_{up}$ with the non-linear term $aQ^b$. This model is less ambiguous. In my personal experience, the coefficients $a$ and $b$ of the non-linear model also give much more insight into the influence of the river discharge on the mean water level along tidal rivers.

Our reply: We very much appreciate the comments raised by the Reviewer. In this case, the regression model can be described by the following equation:

$$Z = Z_0 / \text{std}(Z_0) + \alpha \left[ Q / \text{std}(Q) \right]^{\beta} + \gamma Z_{\text{down}} / \text{std}(Z_{\text{down}}) \qquad \text{(R1)}$$

where the potential influence of $Z_{up}$ on water level dynamics is implicitly accounted by the nonlinear term $\alpha Q^{\beta}$. It can be seen from Figure R2 and Table R1 that the model performance is more or less the same as the original triple linear regression model, except that the RMSE values are slightly larger at NJ, MAS and WH stations (ranging between 0.17 and 0.21 m) than those using the triple linear regression model (ranging between 0.11 and 0.15 m).

In the revised manuscript, we have explicitly mentioned the reason why we include the upstream water level $Z_{up}$ in the regression model: "*To clarify the importance of including $Z_{up}$ in the regression model, we replaced the terms $\alpha Q/std(Q) + \gamma Z_{up}/std(Z_{up})$ with the nonlinear term $\alpha[Q/std(Q)]^{\beta}$ in Equation (1). In this case, the model performance is more or less the same as the original triple linear regression model (see*

[Figure]

Figure R2. Comparison between predicted and observed daily averaged water levels for both the pre-TGD and post-TGD periods at different gauging stations along the YRE by replacing the terms $\alpha Q/\text{std}(Q) + \gamma Z_{up}/\text{std}(Z_{up})$ with the nonlinear term $\alpha[Q/\text{std}(Q)]^{\beta}$ in Equation (1): (a) Jiangyin (JY), (b) Zhenjiang (ZJ), (c) Nanjing (NJ), (d) Maanshan (MAS), (e) Wuhu (WH).

Table R1. Calibrated regression coefficients for both the pre-TGD and post-TGD periods along the YRE by replacing the terms $\alpha Q/\text{std}(Q) + \gamma Z_{up}/\text{std}(Z_{up})$ with the nonlinear term $\alpha[Q/\text{std}(Q)]^{\beta}$ in Equation (1).

| Stations | | $Z_0$ | $\alpha$ | $\beta$ | $\gamma$ | RMSE/m | Standard deviation/m |
|---|---|---|---|---|---|---|---|
| JY | pre-TGD | -0.10 | 0.31 | 0.62 | 0.49 | 0.06 | 0.64 |
| | post-TGD | -0.44 | 0.55 | 0.45 | 0.43 | 0.08 | 0.59 |
| ZJ | pre-TGD | 0.00 | 0.89 | 0.92 | 0.48 | 0.13 | 1.23 |
| | post-TGD | -0.27 | 1.02 | 0.82 | 0.43 | 0.14 | 1.12 |
| NJ | pre-TGD | -0.37 | 1.84 | 0.81 | 0.40 | 0.17 | 1.72 |
| | post-TGD | -0.55 | 1.57 | 0.84 | 0.40 | 0.19 | 1.54 |
| MAS | pre-TGD | -0.57 | 2.50 | 0.77 | 0.38 | 0.20 | 2.02 |
| | post-TGD | -0.64 | 2.03 | 0.83 | 0.37 | 0.21 | 1.80 |
| WH | pre-TGD | -1.31 | 3.78 | 0.66 | 0.32 | 0.21 | 2.35 |
| | post-TGD | -1.36 | 3.09 | 0.71 | 0.32 | 0.21 | 2.07 |

2. The regression model does not include the effect of the tides on the mean water level. However, this effect is not negligible during periods of low river flow (LeBlond, 1978). This introduces a systematic error. As the Three Gorges dam increased river discharge during the low flow season, this can bias the results. It is, therefore, reasonable to include the influence of tides on the mean water level in the regression model. For example, Kukulka and Jay (2003) suggest the regression model inear in $h^3$:

$$h^3 \approx aQ_{river}^2 + b \, | \, z_{tide} \, |^2 + c,$$

while (Kastner et al., 2019) suggested linearizing the backwater equation, which can be readily approximated in a regression model linear in $h$ (or $z$).

Our reply: Actually, the potential effect of the tides on the mean water level is implicitly considered by the $Z_{down}$ term, which is typically featured by a spring-neap cycle. Figure R3 shows the autoregressive power spectral density estimate of the daily averaged water level observed at TSG gauging station, where significant periodic cycle of 14.8 days was observed.

In the revised manuscript, we have explicitly mentioned that: "*It should be noted that the imposed downstream water level $Z_{down}$ also implicitly accounts for other nontidal factors, such as wind, ocean temperature and ocean salinity, which are assumed to be negligible in the regression model when compared with the tidally induced water level fluctuations featured by a typical spring-neap cycle (see Figure S2 in the Supplementary Material)*". (see Lines 130-134)

[Figure]

Figure R3. Autoregressive power spectral density estimate of the daily averaged water level observed at TSG gauging station.

3. The independent variables are not normalized in the regression model so that the coefficients have very different magnitudes ($O(\alpha) = 10^{-5}$ while $O(\beta) = 1$). It is thus not obvious which predictor (downstream or upstream level) has the largest influence at a particular location. This can be revealed by normalizing the independent variables by their standard deviation before the regression:

$$Z = Z_0 + \alpha Q / \mathrm{std}(Q) + \beta Z_{\mathrm{down}} / \mathrm{std}(Z_{\mathrm{down}}) + \gamma Z_{\mathrm{up}} / \mathrm{std}(Z_{\mathrm{up}})$$

This is preferable to the order in the study, where variance is normalized after the regression.

Our reply: We thank the reviewer to point this out. In the revised manuscript, we have normalized the input parameters by their standard deviations.

4. Interpolation of slopes and uncertainty estimates (Figure 4 and 5, lines 190_)
■   There is a mistake in the slope calculation. The values should be in the order $10^{-5}$, not $10^{-8}$. The distance between the stations was probably not converted from km to m.

Our reply: You are right! In the revised manuscript, we have corrected this mistake (see Figure R4 below).

[Figure]

Figure R4. Reconstructed spatial-temporal water levels, Z, (a, c) and their slopes, S, (b, d) for the climatological year during both the pre-TGD (a, b) and post-TGD (c, d)

periods. The red lines in subplots (b) and (d) indicate the local minimum water level slopes in the central section of the YRE (between Jiangyin and Zhenjiang).

- ■ Determining the slope from by higher-order (Hermite) interpolation is not meaningful here. This is because the error (of the slope) is amplified at the interpolated values between the stations. As a consequence, the interpolated slope has unrealistic local extrema at the midpoints between stations (Figure 5). The error of the cubically (Hermite) interpolated slope at the midpoint between stations is about 1.8 times as large as the errors of the levels at the stations. Since the error of the levels is about 10%, the error of the slope is about 20%. The local maxima of the slope, as well as the difference between the pre- and post-TGD period as indicated in Figure 5 are therefore insignificant. The interpolation error (of the slopes) can be considerably reduced by calculating the slopes at the midpoint between two stations and then linearly interpolating the slopes between the midpoints. In this case the error of the slope is only 0.7 times that of the error in levels. If the authors want to retain cubic interpolation, then the spurious extrema can be suppressed by fitting the 4 coefficients of the cubic polynomial with all 5 stations in a least squares manner.

Our reply: Many thanks for the reviewer's comments on the interpolation of the results. Actually, we only interpolated the daily averaged water level along the estuary, while the slope is computed on the basis of the interpolated water level using the Matlab "gradient.m" function. However, it is true that the unrealistic local extrema are mainly due to the amplification of the error of the interpolated water level.

In the revised manuscript, we have explicitly mentioned that: "*Using the calibrated regression models and interpolated linear regression coefficients (see Figure 3), the spatial-temporal water level dynamics for the two study periods can be reconstructed along the upper YRE for the climatological reference year (Figure 4), which is defined by evaluating for each day of the year the average value of all measurements available over the study period for the same day (though February 29th during leap years was not considered). Subsequently, we used the Matlab 'gradient.m' function (returning the one-dimensional numerical gradient of imposed vector) to estimate the residual water level slope based on the reconstructed water levels along the YRE*". (see Lines 219-225)

- ■ As the model is not parsimonious, it might fit well even if the regression coefficient are uncertain, as multiple parameter combinations can result in similar good model performance. A good way to assess the uncertainty is bootstrapping (Efron and Tibshirani, 1994). Simply split the time series into blocks comprising of one month, this reduces the effect of serial correlation. When there are n blocks, randomly choose $\sqrt{n}$ blocks and fit the model. Repeat this a few hundred times. The standard error is simply the standard deviation of the estimated parameters. The standard error of the coefficient, predicted levels and slopes can then be indicated with error bars in Figure 3 and 5. The cubic interpolation results in larger errors at midpoints between sections, so errors bars are best placed there.

Our reply: We very much appreciate the reviewer's comments on the quantification of the model uncertainty. Actually, since we used the default multiple linear regression function "regress.m" in Matlab, it is possible to output a matrix of 95% confidence intervals for the coefficient estimates, which is similar to the adoption of bootstrapping.

In the revised manuscript, we have updated the Figure 3 by including the error bar for each coefficient (see Figure R5 below). Meanwhile, we have moved the discussion with regard to the interpolated coefficients to the Section 4.1. In addition, we have explicitly mentioned that: "*Spatial interpolation of the triple linear regression coefficients was performed by means of piecewise cubic Hermite interpolants (e.g., Matte et al., 2014) in order to correctly reproduce the water level dynamics at arbitrary locations along the estuary. Figure 3 shows the four spatially interpolated model coefficients together with vertical error bar (estimated using the Matlab 'regress.m' function with 95% confidence intervals) along the upper YRE for the pre-TGD and post-TGD periods. Generally, a longitudinal reduction in coefficients (e.g., $Z_0$ and $\beta$ in Figure 3a, c) in the landward direction suggests a weakening effect of these parameters on the total variations in water levels, which corresponds to the external forcing from the seaward end of the estuary. On the contrary, if the coefficients are increased (e.g., $\alpha$ and $\gamma$ in Figure 3b, d), this corresponds to an enhancement from the upstream end. However, we observed an exception from the MAS to WH stations, where the coefficient $\alpha$ was reduced (see Figure 3b), suggesting a switch of the effect of river discharge in the upstream part of the estuary. The standard error presented in Figure 3 represents the standard deviation of the estimated linear regression coefficients, which suggests that the proposed triple linear regression model is robust with limited uncertainty*". (see Lines 204-217)

[Figure]

Figure R5. Interpolated linear regression coefficients $Z_0$ (a), $\alpha$ (b), $\beta$ (c), $\gamma$ (d) with error bar along the YRE (upstream of the Jiangyin gauging station) for both the pre-TGD and post-TGD periods. The vertical error bar was estimated using the Matlab 'regress.m' function with 95% confidence intervals.

**Minor**

■ Title estuary→upper estuary

Our reply: We agree with the reviewer's comment. The title was revised as: "*Quantifying the impacts of the Three Gorges Dam on the spatial-temporal water level dynamics in the upper Yangtze River estuary*"

■ 35 The term "analytical solution" is misleading, as the water level is still determined by (numerically) integrating an initial value problem (eq. 22 in Cai et al. (2016)).

Our reply: In the revised manuscript, we have replaced "analytical solutions" with "*semi-analytical solutions*". In addition, we also included the reference of Kastner et al. (2019).

■ 44 Kukulka and Jay (2003) should be referenced here, as an important regression model for the mean water level of tidal rivers.

Our reply: In the revised manuscript, we have included the reference of Kukulka and Jay (2003).

■ 45 "these methods suggest that water level dynamics in estuaries are highly

nonlinear and nonstationary" This sounds as if water levels in tidal are difficult to analyse and predict, and that looking at tidal cycle/average is a novel idea. However, there is a large amount of publications how water levels can be approximated well on a cycle-by-cycle basis, see the works of the groups of Savenije, Godin, Jay, Hoitink, and Friedrichs.

Our reply: In the revised manuscript, we have revised this sentence as: "*these methods suggest that water level dynamics in estuaries are highly nonlinear and nonstationary owing to complex tide-river interactions*". (see Lines 46-47)

■ 49 The reference to Darcy is dubious. Even if the surface level can be predicted by a linear regression model, it is still turbulent flow (quadratic flow resistance), which is very different from groundwater ow (linear flow resistance).

Our reply: In the revised manuscript, we have removed "similar to Darcy's law for groundwater flow".

■ 89 Mark Gaoqiaoju on the map in Figure 1

Our reply: In the revised manuscript, we have marked Zhongjun instead of Gaoqiaoju gauging station on the Map according the reference of Zhang et al. (2012).

■ 93 "we mainly concentrate on the tide-river dynamics" This is not the case, since, as commented by me before, the tidally induced water level offset is not included in the regression model.

Our reply: In the revised manuscript, to be more specific, we have replaced "tide-river dynamics" with "*water level dynamics*". Actually, since the input parameter $Z_{down}$ implicitly considered the influence induced by the tidal forcing (especially the spring-neap changes), we actually concentrated on the tide-river dynamics.

■ 110 Mention here, which of the stations where chosen as the upstream and the downstream end (Datong and Gaoqiaoju?).

Our reply: We agree with the reviewer's comment. In the revised manuscript, we have explicitly mentioned that "*In this study, the DT hydrological station was chosen as the upstream end, while the TSG gauging station was used as the downstream end.*" (see Lines 129-130)

■ 169 "linear" is misleading here. The water depth in the upstream estuary most likely scale like $h \approx Q^{2/3}$. The non-linearity is just hidden by including $Z_{up}$ in the regression model.

Our reply: In the revised manuscript, we have explicitly mentioned that "*which leads support to our hypothesis that the response of water level dynamics to hydrodynamics at both ends of the estuary is largely linear in the YRE owing to the explicit inclusion of $Z_{up}$ in the regression model.*" (see Lines 194-196)

■ 248 The conclusion "[at the downstream stations] tide dominates [the tidally averaged water level]" sounds odd, as the regression model applied in this study

does not explicitly account for the tidally induced water level offset. It only includes the tidally averaged water level at the seaward station. However, at the river mouth the tidally induced water level offset is negligible as it integrates along the estuary, c.f. Kastner et al. (2019) and Cai et al. (2016). So, no meaningful conclusion about the tidal influence can be drawn. The model probably indicates that fluctuations of the sea level unrelated to tides, such as wind, ocean-temperature and ocean-salinity, dominate the mean water level dynamics near the sea. It would be insightful to actually determine the tidal influence by including it explicitly in the regression model.

Our reply: Actually, since the input parameter $Z_{down}$ in the regression model implicitly considered the influence induced by the tidal forcing (especially the spring-neap changes), we actually concentrated on the tide-river dynamics.

In the revised manuscript, we have explicitly mentioned that: "*It should be noted that the imposed downstream water level $Z_{down}$ also implicitly accounts for other nontidal factors, such as wind, ocean temperature and ocean salinity, which are assumed to be negligible in the regression model when compared with the tidally induced water level fluctuations featured by a typical spring-neap cycle (see Figure S2 in the Supplementary Material)*". (see Lines 130-134)

■ 248 The river discharge influences the salinity gradient, and with it the variation of the water level at the reference station at the sea (Savenije, 2012). The influence on river discharge on the downstream stations might thus be larger than indicated by the model.

Our reply: We agree with the reviewer that the salinity gradient may influence the water level at the reference station at the sea. However, since the study area is out of the maximum salt intrusion length, thus the potential influence due to salinity gradient is negligible.

■ 257 This paper has →We have

Our reply: In the revised manuscript, we shall replace "This paper has" with "*In this study, we have*". (see Line 294)

■ 263 It was shown →We show

Our reply: We agree with the reviewer's comment.

■ 271 How relevant are (seasonal) changes of roughness and bedforms, due to changes in water and sediment supply by the dam?

Our reply: Here we can conclude that the main impact due to changes in water and sediment supply by the dam tends to deepen the riverbed since the alterations caused by geometric changes are negative.

■ Figure 2 It would be more meaningful to plot ($z_{pred}$ - $z_{obs}$) vs $z_{obs}$ and to use smaller dots which do not overlap that much. This would reveal better any systematic

variation.

Our reply: We agree with the reviewer's comments. In the revised manuscript, the Figure 2 was revised as follows (see Figure R6 below).

[Figure]

Figure R6. Alterations in difference between predicted and observed daily averaged water levels as a function of observed daily averaged water levels for both the pre-TGD and post-TGD periods at different gauging stations along the YRE: (a) Jiangyin (JY), (b) Zhenjiang (ZJ), (c) Nanjing (NJ), (d) Maanshan (MAS), (e) Wuhu (WH).

- Figure 3 Add subplots titles, like Discharge, Downstream level, Upstream level so that the figure can be interpreted without looking up the meaning of the coefficients $\alpha$, $\beta$, $\gamma$.

Our reply: We agree with the reviewer's comment. In the revised manuscript, we have included the subplots titles (see Figure R5 above).

- Figure 3 begins from Jiangyin→upstream of Jiangyin

Our reply: We agree with the reviewer's comment.

- Figure 7 The average annual average hydrograph of the post-TGD period is corrupted by high-frequent fluctuations of the hydrograph. The graph would be clearer if the fluctuation is removed it through by smoothing with a sliding window. A triangular window with a width of 30 days seems appropriate. Smooth the data for the pre-TGD period as well, for better comparison.

Our reply: We agree with the reviewer's comments. In the revised manuscript, the

Figure 7 was revised as follows (see Figure R7 below).

Figure R7. Alterations in river discharge and water level observed at DT and TSG, respectively, during the post-TGD period relative to the pre-TGD period over the climatological year. The daily averaged river discharge and water level were smoothed using a moving average filter with a span of 30 days.

**Response letter to Reviewer#2**

**Comments:**

In this study, the authors investigated the spatial-temporal water level dynamics along the main stream of the Yangtze River estuary by means of a triple linear regression model accounting for both the upstream and downstream boundary conditions. The model was subsequently used to quantify the influence of the Three Gorge Dam's operation on the water level dynamics. Results showed that the alteration in water level dynamics are mainly controlled by the variation in freshwater discharge owing to the Three Gorge Dam's operation, while the influence by geometric changes are minor when compared with that of the river discharge alteration. The first reviewer already provided many constructive comments on the manuscript, which I mostly agreed, especially concerning the validity of the proposed triple linear regression model. Generally, the paper is well organized and written. However, there are still some concerns which should be properly addressed before the paper can be accepted in the Ocean Science.

Our reply: We very much appreciate all the comments and suggestions raised by the

reviewer. In the revised manuscript, we have completely addressed all the comments.

**Major concerns:**

1. The authors assumed that the alteration in water level dynamics can be primarily attributed to the geometric change (caused by the combined influences of both natural and anthropogenic modifications) and the boundary effects (induced by the changes in upstream and downstream conditions, primarily due to the TGD's freshwater regulation). Since the authors proposed a triple linear regression model to quantify the impacts of the Three Gorges Dam (representing the intensive human intervention) on the water level dynamics, how did the authors account for the potential impacts due to the climate change (such as intensifying precipitation, global sea level rise, etc.)?

Our reply: We thank the reviewer for pointing this out. Indeed, for the time being, we assumed that the largest contribution to the alteration of river discharge before and after the TGD can be primarily attributed to the TGD's freshwater regulation, which is not completely true due to the influences of other dams (such as Gezhouba dam) and the climate change (such as intensifying precipitation over the river basin). Similarly, the potential influence of climate change (such as global sea level rise) may slightly alter the water level at the downstream boundary. Consequently, in the revised manuscript, we have clarified that: "*It is worth noting that the quantity $\Delta_{BOU}$ (including both the upstream and downstream boundary conditions) should be interpreted as the water level alteration owing to the overall influences driven by both human interventions and climate change. However, in this study the largest contribution to the alteration in upstream boundary condition (i.e., river discharge) can be primarily attributed to the TGD's operation, since the TGD alone accounts for more than 30% of the total storage capacity of the dams constructed between 1987 and 2014 along the Yangtze River (Li et al., 2016). In addition, we note that the only other dam (Gezhouba, abbreviated by GZB, see Figure 1a) along the main course of the Yangtze River was constructed in 1981 (before the TGD). With regard to the downstream boundary condition, the adopted water levels observed at TSG station implicitly account for the potential impacts induced by both anthropogenic (such as channel dredging) and climate (such as global sea level rise) changes.*" (see Lines 170-180)

2. It was mentioned by the authors that the proposed model is particularly useful for determining scientific strategies for sustainable water resources management in dam-controlled estuaries worldwide. Actually, as far as I see, the proposed method can also be used to quantify the influence of climate change on spatial-temporal water level dynamics since both the upstream and downstream boundary conditions are closely related to the climate change even without the construction of large dams. Further comments with regard to the applicability of the proposed method can be clarified.

Our reply: We agree with the reviewer's comment. In the revised manuscript, we have clarified that: "*Such a novel approach should be particularly helpful for determining scientific guidelines for sustainable water resources management (e.g., dredging for navigation, flood control, salt intrusion prevention etc.) in estuaries worldwide, especially for dam-controlled estuaries. In addition, the proposed method can also be*

*used to quantify the potential impacts of changes in boundary conditions induced by climate change (such as intensifying precipitation, global sea level rise, etc.) in natural estuaries without considerable human interventions*". (see Lines 343-349)

In addition, we have slightly modified the last sentence in the abstract part: "*The presented method to quantify the separate contributions made by changes in boundary conditions and geometry on spatial-temporal water level dynamics is particularly useful for determining scientific strategies for sustainable water resources management in dam-controlled or climate-driven estuaries worldwide*". (see Lines 17-20)

3. The geometric effect in this paper is mainly referred to the bathymetric changes in the estuarine system, which should be the primary factor dominating the geomorphological changes in the Yangtze river estuary. However, for other estuarine systems, the geometric effect could also due to the lateral boundary changes. Could the authors give some comments on the applicability of the proposed method to such cases?

Our reply: In the revised manuscript, we have clarified that: "*Meanwhile, it is also worth noting that the quantity $\Delta_{GEO}$ should be interpreted as the water level alteration due to the overall impacts caused by both the bathymetric change and the storage area change.*" (see Lines 180-182)

4. Finally, I would suggest the authors to clarify the implications of this contribution.

Our reply: We very much appreciate this suggestion raised by the reviewer. In the revised manuscript, we have explicitly mentioned that: " *There exists a long tradition of statistical, analytical and numerical studies on tide-river interactions in estuaries worldwide, such as the Columbia River estuary in the USA (e.g., Kukulka and Jay, 2003; Jay et al., 2015; Pan et al., 2018b), the St. Lawrence River estuary in Canada (e.g., Godin,1999; Matte et al., 2013, 2014), the Mahakam River estuary in Indonesia (e.g., Buschman et al., 2009; Sassi and Hoitink, 2013), the Yangtze River estuary in eastern China (e.g., Guo et al., 2015, 2020; Yu et al., 2020) and the Pearl River estuary in southern China (e.g., Zhang et al., 2018; Cai et al., 2018b, 2019b). These studies showed that as tides propagate along the estuary the tidal amplitude, phase and shape were influenced by the bottom friction, channel geometry and river discharge. In this study, with the proposed simple yet effective triple linear regression model, we are able to isolate and to quantify the impacts of the boundary (such as freshwater regulation due to dam's operation) and geometric (such as channel dredging) effects on the tide-river dynamics. Such a novel approach should be particularly helpful for determining scientific guidelines for sustainable water resources management (e.g., dredging for navigation, flood control, salt intrusion prevention etc.) in estuaries worldwide, especially for dam-controlled estuaries. In addition, the proposed method can also be used to quantify the potential impacts of changes in boundary conditions induced by climate change (such as intensifying precipitation, global sea level rise, etc.) in natural estuaries without considerable human interventions.*" (see Lines 333-349)

[revised manuscript text omitted]

Unravelling the causes of tidal asymmetry in deltas, J. Hydrol., 564, 588–604.

https://doi.org/10.1016/j.jhydrol.2018.07.023, 2018.

---

## Author Response (AR3)

**Response letter**

We thank the Editor and the Reviewers for the careful consideration of our work. Their constructive and thoughtful comments and suggestions led to a much improved and complete revision of the manuscript. In the revised paper, we have addressed all the comments formulated by the Reviewers by replying (in black) to their remarks (in blue). The lines numbers in this rebuttal refer to the revised version of the manuscript.

**Editor's general comments**

**Comments:**
I now have both referees' comments on the latest version of your manuscript. They are favourable overall but there are some "Referee comments" which I copy below, as well as some of my own, for you to address in a revised manuscript please.
Our reply: We very much appreciate all the comments and suggestions raised by the editor and reviewers. In the revised manuscript, we have completely addressed all the comments.

**Responses to Reviewer#1's comments**

■ 2.2 Datasets. In the study both discharge and water level at the upstream station is used, which implies that there is are continuous measurements of both the stage and flow velocity. Yet measurements methods can change over time, and even if they do not change, they require frequent recalibration to account for morphological changes at the gauging station. Therefore, it would be insightful to provide some information on how discharge at the Datong station is measured, if the method of measurement changed during the study period, and most importantly, if it was regularly updated to account for scouring of the bed, after the TGD had been constructed.

Our reply: In the revised manuscript, we have explicitly mentioned that: "*Here, it is worth noting that the observed river discharges at the DT hydrological station were generally derived from well-calibrated stage-discharge relationship, which is established by concurrent measurements of stage and discharge (through approximately 50-70 filed measurements of flow depth and velocity in each year to account for the cross section changes) over a wide range of river discharge conditions.*" (see Lines 98-102)

■ 116-117 standard deviation function → standard deviation

Our reply: Corrected as suggested.

- 120-121 "daily averaged water levels observed at the DT hydrological station are not uniform for identical river discharge" → "There is no unique stage-discharge relation at the Datong hydrological station"

Our reply: Corrected as suggested.

- 121-122 "due to the influence of external forcing [...]". A potentially important factor, the stage-discharge hysteresis, is not mentioned. Is it not relevant at Datong? I suggest to provide a rough estimate of the stage-discharge hysteresis.

Our reply: We thank the reviewer to point this out. Actually, the stage-discharge hysteresis effect is a key factor leading to a non-unique stage-discharge relationship. In the revised manuscript, we have explicitly mentioned that: "*It is worth noting that there is no unique stage-discharge relationship at the DT hydrological station (see Figure S1 in the Supplementary Material) owing to the stage-discharge hysteresis effect caused by flow unsteadiness, together with the influence of external forcing, either the potential influence induced by the tidal forcing (especially during the dry season) or the exerted residual water level slope upstream of the DT hydrological station (owing to the relative importance of river discharge between the main stream and the tributaries, especially during the flood season).*" (see Lines 125-131)

- 139 variance function → variance

Our reply: Corrected as suggested.

- 176 Note that Gezhouba is also a run-of-the-river dam, and therefore should not considerably influence the discharge regime.

Our reply: We thank the reviewer to point this out. Indeed, Gezhouba is also a run-of-the-river dam. In the revised manuscript, we have explicitly mentioned that: "*In addition, we note that the only other dam (Gezhouba, abbreviated by GZB, see Figure 1a) along the main course of the Yangtze River was constructed in 1981 (before the TGD) and should not considerably influence the discharge regime since it is a run-of-the-river hydroelectric system.*" (see Lines 179-182)

- 212 increased → increasing

Our reply: Corrected as suggested.

- 215 "The standard error [...] represents the standard deviation" → "The error-bars [...] represent the standard error"; The standard error and standard deviation are related but not identical (serr $\propto$ sd/√nsample).

Our reply: In the revised manuscript, we have revised this sentence as: "*The error bars presented in Figure 3 represent the standard deviation of the estimated linear*

*regression coefficients, which suggests that the proposed triple linear regression model is fitting well.*" (see Lines 219-221)

■ 217 is robust → is fitting well. ["Robust" in statistics implies that a method to suppress outliers was employed, which is not the case here.

Our reply: Corrected as suggested.

■ 215-217 "[...] the standard error [...] suggests that the proposed triple linear regression model is [fitting well] with limited uncertainty". Remove the qualifier "with limited uncertainty", as a good fit does not imply low uncertainty. In general, the goodness of fit to the measured values improves when more parameters are added, but the reliability of predicting values at moments for which no measurements are available decreases (overfit). (See also my recommendation in the previous revision to validate the model through bootstrapping.)

Our reply: In the revised manuscript, we have revised this sentence as: "*The error bars presented in Figure 3 represent the standard deviation of the estimated linear regression coefficients, which suggests that the proposed triple linear regression model is fitting well.*" (see Lines 219-221)

■ 312 constant value of local mean sea level → constant mean sea level

Our reply: Corrected as suggested.

■ 310-313 "[The] channel deepening [...] tend[s] to increase in the landward direction [..]. This phenomenon can be primarily attributed to the constant value of local mean sea level or the ultimate base level that the topography tends to approach due to erosion."
I cannot follow this argument, as the constant sea level in combination with the seasonal discharge variation promotes, not prevents, scouring c.f. theoretical work by (Lamb et al., 2012) for the Mississippi and measured longitudinal river profiles of the Mahakam (Sassi et al., 2012) and Kapuas (Kästner et al., 2017). I propose two alternative hypotheses: First, reduced sediment supply initially just results in scouring downstream in the vicinity of the dam, after which the scour slowly propagates further downstream with time. Second, the reduction of seasonal discharge variation by the TGD reduces the overdeepening near the sea.

Our reply: We thank the reviewer to point this out. Based on the phenomenon that the geometric changes $\Delta_{GEO}$ (mainly caused by channel deepening) tend to increase in the landward direction (see Figure 6c in the manuscript), we agree with the hypotheses proposed by the reviewer. In the revised manuscript, we have explicitly mentioned that: "*In addition, this phenomenon is also closely related to the scouring downstream near the TGD, which slowly propagates further downstream due to the reduced sediment supply (see also Lamb et al., 2012; Sassi et al., 2012; Kästner et al., 2017). Moreover, the reduction of seasonal discharge variation due to TGD's regulation may probably reduce the overdeepening near the sea.*" (See Lines 322-326)

■ 323-328: Since the TGD continues to deprive the Yangtze of sediment, it is reasonable to assume that the scouring will continue. Can the authors hypothesize how the water levels will evolve in future?

This also points to a potential methodological limitation of the study, as the mean conditions are treated as if they were stationary before and after the dam construction, while the geometric influence has likely gradually increased since construction of the dam due to ongoing scouring.

Our reply: We thank the reviewer to point this out. Indeed, we agree with the reviewer that the souring will continue owing to the sediment trapping effect due to TGD's operation. And we also agree that the assumption of stationary condition before and after the TGD is one of the model limitations. Thus, in the revised manuscript, we have explicitly mentioned that: "*It is also worth noting that in this study we assumed a more or less stationary condition before and after the TGD's construction for the regression model, which is not completely true due to the gradually increased geometric influence caused by the TGD.*" (See Lines 338-341)

**Responses to editor's comments**

■ 125. "extern" → "external"

Our reply: Corrected as suggested.

■ 129-130. "In this study, the DT hydrological station was chosen as the upstream end, while the TSG gauging station was used as the downstream end." I think this sentence should be moved to before the present line 120 where DT is referred to but the reader does not presently know that it gives Zup.

Our reply: We thank the editor to point this out. In the revised manuscript, we have moved the sentence "*In this study, the DT hydrological station was chosen as the upstream end, while the TSG gauging station was used as the downstream end*" before the present line of 120.

■ 186-188. Not a sentence – no verb! Maybe (line 186) ". . (see Figure 2) for . ." although this makes a long sentence.

Our reply: We thank the reviewer to point this out. In the revised manuscript, we have revised the sentence as: "*The proposed triple linear regression model was applied to reproduce the water level dynamics observed during both the pre-TGD and post-TGD periods for the given upstream river discharges and water levels observed at the DT hydrological station and the water levels observed at the TSG gauging station(see Figure 2).*" (see Lines 190-193)

■ 192. Omit "accounting for" which tends to suggest that the model only accounts for 4-13% of the standard deviations.

Our reply: Corrected as suggested.

■ 207-208. "(estimated using the Matlab 'regress.m' function with 95% confidence

intervals)" is unnecessary, it repeats the figure 3 caption.

Equation (3) tends to identify "tidal" Pt with Zdown. Indeed Zdown will include tides, but it will also include effects of "wind, ocean-temperature and ocean-salinity" (Reviewer comment on earlier manuscript). The last paragraph of section 4 should discuss these other influences.

Our reply: We thank the editor to point this out. In the revised manuscript, we have deleted the sentence: "*(estimated using the Matlab 'regress.m' function with 95% confidence intervals)*". In addition, in the previous manuscript, we have explicitly mentioned that: "*It should be noted that the imposed downstream water level $Z_{down}$ also implicitly accounts for other nontidal factors, such as wind, ocean temperature and ocean salinity, which are assumed to be negligible in the regression model when compared with the tidally induced water level fluctuations featured by a typical spring-neap cycle (see Figure S2 in the Supplementary Material).*" (see Lines 135-139)

- 257 232-239. I think the Reviewer comment (on the earlier manuscript) about errors in estimated slopes is still relevant. See the reviewer comment above about lines 215-217; the figure 5 plots of slopes show signs of over-fitting. Anyway, how does the Matlab "gradient.m" function estimate / interpolate slopes?

Our reply: We thank the editor to point this out. Actually, we interpolated the reconstructed water levels along the channel from JY to WH with the interval being 1 km using the cubic spline interpolation. Subsequently, the water level slope can be derived by calculating the slope between the adjacent points along the channel, which is done by the Matlab "gradient.m" function. In the revised manuscript, we have explicitly mentioned that: "*Subsequently, we used the Matlab 'gradient.m' function (i.e., 'gradient' calculates the central difference for interior data points, while it calculates values along the edges of the matrix with single-sided differences, see details in https://www.mathworks.com/help/matlab/ref/gradient.html) to estimate the residual water level slope based on the reconstructed water levels along the YRE.*" (see Lines 227-231)

- 315-316 and figure 9. The text and figure agree but Z1 as original and Z0 as new is confusing especially as they give $\Delta Z$ = - change in Z.

Our reply: In the revised manuscript, we have used the absolute value $|\Delta Z|=|Z_0-Z_1|$ instead of $\Delta Z= Z_1-Z_0$ to avoid any confusing understanding.

References

Kästner, K., A. J. F. Hoitink, B. Vermeulen, T. J. Geertsema, and N. S. Ningsih, Distributary channels in the fluvial to tidal transition zone, Journal of Geophysical Research: Earth Surface, 122, 696–710, 2017.

Lamb, M. P., J. A. Nittrouer, D. Mohrig, and J. Shaw, Backwater and river plume controls on scour upstream of river mouths: Implications for fluvio-deltaic morphodynamics, Journal of Geophysical Research: Earth Surface, 117, F01002, 2012.

Sassi, M. G., A. J. F. Hoitink, B. Brye, and E. Deleersnijder, Downstream hydraulic geometry of a tidally influenced river delta, Journal of Geophysical Research: Earth

Surface, 117, F04022, 2012.

---

## Author Response (AR4)

**Response letter**

We thank the Editor for the careful consideration of our work. In the revised paper, we have addressed all the comments formulated by the Editor by replying (in black) to their remarks (in blue). The lines numbers in this rebuttal refer to the revised version of the manuscript.

**Editor's comments**

**Comments to the author:**

Thank-you for your (re-)revised manuscript. I think there are still a few points that need clarification; please see specific comments below. Please also remember that on final publication all these comments and responses will be available to readers who will be able to see whether comments have been responded to appropriately. Accordingly I am asking for Minor Modifications.

Our reply: We very much appreciate all the comments and suggestions raised by the editor. In the revised manuscript, we have completely addressed all the comments.

**Specific comments**

Line 202. Please either omit "," before "and an increase" or add "," after "parameter" in line 203. ["after the construction of the TGD" in line 203 applies to all the parameters, not just $\gamma$.]

Our reply: You are right! In the revised manuscript, we have add "," after "parameter" in Line 203 (see Line 203).

Lines 219-220. The text now reads ". . The error bars in Figure 3 represent the standard deviation of the estimated linear regression coefficients . ." Do you mean "standard deviation" or "standard error"? The reviewer suggested "standard error". "standard error" seems more likely to me because I suppose that: you only have one estimated value for each regression coefficient; there is no standard deviation which would need many values; you are estimating the uncertainty (possible error) of the coefficient from the residual error in the regression.

Our reply: Yes, it should be "standard error". In the revised manuscript, we have corrected this mistake. (see Line 220)

Lines 268-270. "while it remained more or less constant during May to June (increasing slightly by 0.01 m), and it generally decreases during the rest of the year by approximately 0.54 m" does not correspond with the appearance of figure 6a. There is much variation during May and June (more than earlier in the year) and 0.54 m must be some sort of average of values that differ greatly with time and location. In line 310 you have 0.46 m

not 0.54 m.

Our reply: Actually, these values are the monthly averaged alterations, which are presented in Table 2. In the revised manuscript, we have explicitly mentioned that: "*From January to March, the total alteration $\Delta_{TOT}$ increased by approximately 0.28 m* **on a monthly scale over five different gauging stations along the upper YRE**, *while it remained more or less constant during May to June (increasing slightly by 0.01 m), and it generally decreases during the rest of the year by approximately 0.54 m (**see Figure 6a and Table 2**)*" (see Lines 270-274).

In addition, we have corrected the values of alterations in Lines 309-310.

Lines 340-341. Please explain "due to the gradually increased geometric influence caused by the TGD" by reference to "ongoing scouring".

Our reply: In the revised manuscript, we have explicitly mentioned that: "*It is also worth noting that in this study we assumed a more or less stationary condition before and after the TGD's construction for the regression model, which is not completely true due to the gradually increased geometric influence (such as ongoing scouring) caused by the TGD (e.g., Yang et al., 2022)*". (see Lines 345-348)

In lines 135-139 you write "It should be noted that the imposed downstream water level $Z_{down}$ also implicitly accounts for other nontidal factors, such as wind, ocean temperature and ocean salinity, which are assumed to be negligible in the regression model when compared with the tidally induced water level fluctuations featured by a typical spring-neap cycle (see Figure S2 in the Supplementary Material)." I think you should follow this up in the last paragraph of the Discussion section 4; are these other factors really negligible? [How much uncertainty do they introduce to the regression?]

Our reply: Since the potential impacts induced by nontidal factors (such as wind, ocean temperature and ocean salinity) on the regression model are not trivial things, thus we suggest that further study is needed in the manuscript. In the last paragraph of the Discussion section 4, we have explicitly mentioned that: "*Here, it should be noted that the contribution $p_t$ implicitly accounts for both tidal and nontidal factors (e.g., wind, ocean temperature and ocean salinity), hence further study is required to quantify the potential influences due to nontidal factors*". (see Lines 303-305)

Figure 5. From your present text I infer that the plotted slopes are quadratics deriving from the cubic splines used in the interpolation of elevation between gauging stations. Clearly the splines give continuous values of elevation and slope at each station; however, is there any basis for the variation of slope between stations? The plots of slopes look like over-fitting. Could you get continuous values of elevation and slope at each station using only quadratic splines for elevation?

Our reply: We very much appreciate the comments raised by the editor. Actually, the over-fitting of water level slopes is closely related to the cubic spline interpolation of the linear regression coefficients along the upper YRE (see Figure 3 in the manuscript) since the elevations are directly computed using the triple linear regression model. However, since only the first derivatives of two quadratic splines are continuous at the

interior points, the performance of quadratic spline interpolation is generally not as good as the cubic spline interpolation. As we can see from Figure R1 below, the interpolated curves for both $Z_0$ and $\alpha$ are much more fluctuant when compared with those using cubic spline interpolation. Thus, we would prefer to adopting the cubic spline method. In the revised manuscript, we have explicitly mentioned that: "However, cautions should be taken when interpreting each spline going through two consecutive observed data points owing to the overfitting of the linear regression coefficients using the cubic spline interpolation method". (see Lines 246-248)

[Figure]

Figure R1. Interpolated linear regression coefficients $Z_0$ (a), $\alpha$ (b), $\beta$ (c), $\gamma$ (d) using the quadratic spline interpolation with error bar along the upper YRE (upstream of the Jiangyin gauging station) for both the pre-TGD and post-TGD periods. The vertical error bar was estimated using the Matlab 'regress.m' function with 95% confidence intervals.

Figure 6. The labels at the top and the caption suggest that the thick blue curve is $\Delta$TOT at DT in (a), $\Delta$BOU at DT in (b). I think you want "DT $\Delta$Q" –> "$\Delta$Q at DT" in the labels at the top.
Our reply: Corrected as suggested (see Figure R2 below).

[Figure]

Figure R2. Alterations in water levels induced by the combined impacts of natural and anthropogenic changes $\Delta_{TOT}$ (a), boundary condition changes $\Delta_{BOU}$ (b), and geometric changes $\Delta_{GEO}$ (c) at different gauging stations along the upper YRE.

Lines 321-322 and figure 9. I am happy with |ΔZ| etc. but Z1 as original and Z0 as new is still strange; usually 0 comes before 1.
Our reply: In the revised manuscript, we have updated this figure by defining $Z_1$ as the new profile, while $Z_0$ being the original profile (see Figure R3 below).

[Figure]

Figure R3. Illustration of the effect of riverbed deepening on the water level dynamics along the channel.

References

Yang, Y., Zheng, J., Zhu, L., Zhang, H., Wang, J., Influence of the Three Gorges Dam on the transport and sorting of coarse and fine sediments downstream of the dam, Journal of Hydrology (2022), doi: https://doi.org/10.1016/j.jhydrol.2022.128654.